# Polerovirus N-terminal readthrough domain structures reveal molecular strategies for mitigating virus transmission by aphids

Carl J. Schiltz [1,6,10], Jennifer R. Wilson [2,7,10], Christopher J. Hosford[1,8], Myfanwy C. Adams[1], Stephanie E. Preising[2], Stacy L. DeBlasio[2,3], Hannah J. MacLeod[3,9], Joyce Van Eck[4,5], Michelle L. Heck [2,3,5] ✉ & Joshua S. Chappie [1] ✉

Poleroviruses, enamoviruses, and luteoviruses are icosahedral, positive sense RNA viruses that cause economically important diseases in food and fiber crops. They are transmitted by phloem-feeding aphids in a circulative manner that involves the movement across and within insect tissues. The N-terminal portion of the viral readthrough domain (NRTD) has been implicated as a key determinant of aphid transmission in each of these genera. Here, we report crystal structures of the NRTDs from the poleroviruses turnip yellow virus (TuYV) and potato leafroll virus (PLRV) at 1.53-Å and 2.22-Å resolution, respectively. These adopt a two-domain arrangement with a unique inter-digitated topology and form highly conserved dimers that are stabilized by a C-terminal peptide that is critical for proper folding. We demonstrate that the PLRV NRTD can act as an inhibitor of virus transmission and identify NRTD mutant variants that are lethal to aphids. Sequence conservation argues that enamovirus and luteovirus NRTDs will follow the same structural blueprint, which affords a biological approach to block the spread of these agricultural pathogens in a generalizable manner.

Poleroviruses (Family: *Solemoviridae*), enamoviruses (Family: *Solemoviridae*) and luteoviruses (Family: *Tombusviridae*), formerly described as luteovirids but herein referred to as P/E/L viruses, are insect vector-borne, icosahedral viruses capable of infecting most major crop and biofuel plants. Their positive-sense RNA viral genomes are roughly 5.8 kb in size and share a conserved arrangement of open reading frames that spawn five to nine known gene products[1] (Fig. S1a). These in turn orchestrate plant infection and insect transmission through a series of temporally and spatially regulated protein interactions[2]. P/E/L viruses are transmitted almost exclusively by sap-feeding aphid vectors[1,3]. P/E/L virions circulate throughout the aphid body, interacting with proteins in the aphid's gut and accessory salivary glands prior to transmission to a new host plant. The aphid gut represents the first barrier for transmission, providing selectivity for the uptake of P/E/L viruses. Virus replication is limited to the plant phloem and no replication occurs in the insect vector[4]. This mode of transmission is deemed circulative, non-propagative.

[1]Department of Molecular Medicine, Cornell University, Ithaca, NY 14853, USA. [2]Section of Plant Pathology and Plant-Microbe Biology, School of Integrative Plant Sciences, Cornell University, Ithaca, NY 14853, USA. [3]USDA-Agricultural Research Service, Emerging Pest and Pathogen Research Unit, Ithaca, NY 14853, USA. [4]Section of Plant Breeding and Genetics, School of Integrative Plant Sciences, Cornell University, Ithaca, NY 14853, USA. [5]Boyce Thompson Institute for Plant Research, Ithaca, NY 14853, USA. [6]Present address: Department of Biological Sciences, Vanderbilt University, Nashville, TN 37232, USA. [7]Present address: USDA-Agricultural Research Service, Corn, Soybean & Wheat Quality Research Unit, Wooster, OH 44691, USA. [8]Present address: LifeMine Therapeutics, Cambridge, MA 02140, USA. [9]Present address: Accelevir Diagnostics, Baltimore, MD 21202, USA. [10]These authors contributed equally: Carl J. Schiltz, Jennifer R. Wilson. ✉e-mail: mlc68@cornell.edu; chappie@cornell.edu

P/E/L viruses encode two structural proteins[5] (Fig. S1a). The coat protein (CP), derived from ORF3, constitutes the major component of the viral capsid[6]. Stochastic ribosomal readthrough of the CP stop codon generates a second minor capsid component termed the readthrough protein (RTP), which contains an additional readthrough domain (RTD) encoded by ORF5 that is fused to the CP C-terminus[6–9]. The leakiness of the CP stop codon has been maintained throughout evolution and ensures that the RTP is incorporated into the capsid sub-stoichiometrically[10]: mutant viruses that lack the stop codon and make only the full-length RTP cannot assemble proper virions, infect plants, or be transmitted by aphid vectors[11–13]. A soluble form of the RTP that is not associated with the capsid plays a role in phloem limitation and movement within the plant host[13,14]. The readthrough domain itself can be subdivided into a globular N-terminal portion ($^N$RTD) and an unstructured C-terminal portion ($^C$RTD) that undergoes proteolytic processing as part of the normal viral lifecycle[15,16]. Mutant viruses lacking the RTD are not aphid transmissible but form functional capsids capable of protecting the RNA genome and can infect plants at a reduced titer[11,16–18]. In contrast, engineered RTP truncations that remove only the $^C$RTD incorporate efficiently into virions[13], retain the ability to interact with aphid proteins[19] and can be transmitted to new hosts[10]. These observations implicate the $^N$RTD as a key determinant of P/E/L virus transmission and necessary for traversing aphid gut epithelial cells during viral uptake.

Previous structural studies have detailed the underlying organization of P/E/L capsids. Cryo-electron microscopy (cryo-EM) and crystallographic characterization of polerovirus CP constructs lacking the RTD confirmed that P/E/L viruses assemble with T = 3 icosahedral symmetry[20,21], which arranges 180 quasiequivalent monomers into closed particles that display two-fold, three-fold, and five-fold symmetry[22,23] (Fig. S2a, b). Despite these efforts, nothing is known about the structure of the RTD, how it is presented on the capsid, and why it is limited within the mature virion. Deciphering these details is essential for understanding how P/E/L viruses interact with and are transmitted by their aphid vectors.

Here we present atomic-resolution crystal structures of polerovirus $^N$RTDs, which define a two-domain architecture with a unique, interdigitated topology. Our structures rationalize phenotypes observed in previous mutagenesis studies and provide new insights into the organization of the RTD on the capsid surface and the factors limiting its incorporation within mature virions. We also uncover an unexpected evolutionary connection to non-aphid transmissible tombusviruses, which informs how the presence or absence of specific structural features correlate with different requirements for transmission to a new host. Functional experiments establish that the PLRV $^N$RTD can act as an inhibitor of virus transmission and identify $^N$RTD mutant variants that function as potent bioinsecticides. We demonstrate several effective methods for delivering the $^N$RTD to aphids, paving the way for a generalized management strategy to prevent the spread of destructive P/E/L pathogens. Molecular approaches to block virus transmission are of major interest for the development of novel disease control technologies[2] and our findings represent a significant advance toward achieving this in an agricultural setting.

## Results & discussion

### Structural organization of polerovirus $^N$RTDs provides insights into the evolution of aphid transmission

To understand the molecular interactions regulating polerovirus acquisition and transmission by aphids, we generated soluble versions of the $^N$RTD regions from PLRV (residues 230-458 of the complete RTP fusion) and TuYV (residues 224-459) that could be expressed in *E. coli* and purified on the milligram scale for structural and biochemical studies (Fig. S1b, c). Size exclusion chromatography coupled to multi-angle light scattering (SEC-MALS) shows that these constructs form stable dimers in solution (Fig. S1d, e). Both readily crystallized, and we

solved the structure of the PLRV $^N$RTD at 2.22 Å by single wavelength anomalous diffraction (SAD) phasing[24] using selenomethionine-labeled protein (Fig. S3a, b and Table S1). The TuYV $^N$RTD structure was subsequently solved by molecular replacement[25], yielding a more complete model that was refined to 1.53-Å resolution (Fig. 1 and Table S1).

Each TuYV $^N$RTD monomer folds into a two-domain protein comprised of a total of 16 β-strands. Eleven of these strands form two anti-parallel β-sheets−ordered β12-13-7-16-1-4-5 (sheet 1) and β5-6-14-10 (sheet 2)−that sandwich together into a jellyroll fold (Fig. 1, orange). β5 adopts a twisted conformation that runs orthogonal to the plane of the sandwich and connects the two sheets along one edge (Fig. 1a). The short β11 strand connects the sheets on the opposite edge. The additional five strands form an anti-parallel β-sheet (β9-8-15-2-3) that curves into a small barrel with a short a helix (α1) flanking the edge of β3 (Fig. 1, purple). A series of well resolved loops (L1-L5) connect these segments, with L4 folding over and acting as a lid. We designate this barrel the 'cap domain', as it sits above the jellyroll base. The DALI alignment algorithm[26] indicates that the cap domain barrel is present in several unrelated proteins, including the dimerization domains of amino-peptidases, the N-terminal region of the F1-ATPase rotary subunits, the *Aeropyrum pernix* IF5B initiation factor, and mammalian *Norovirus* spike proteins (Fig. S4). Though the topology differs within these proteins, they each maintain a spatially conserved fold (Fig. S4b, c). The TuYV cap domain, however, remains an outlier among this group in that its secondary structure elements are not contiguous, being interspersed throughout the jellyroll domain rather than being connected sequentially to form an isolated globular unit (Fig. 1b and S4). PLRV $^N$RTD monomers adopt the same specific domain arrangement and topology (Fig. S3a, b), suggesting this unique organization is a characteristic feature of poleroviruses and, due to sequence conservation in this region, likely enamo- and luteoviruses as well (Fig. S5).

Further structural comparison using DALI[26] reveals that the jelly-roll folds of the PLRV and TuYV $^N$RTDs are structurally related to the P domains of tombusviruses, with the nearest structural homologs being tomato bushy stunt virus (TBSV) and cucumber necrosis virus. Tombusviruses share key biological properties with P/E/L viruses−including a small, positive-sense single-stranded RNA genome, similar host range, and a non-enveloped icosahedral capsid with T = 3 symmetry comprised of 180 copies of the CP (Fig. S2a, c)−but are distinct in that they are not aphid transmissible and lack an analogous readthrough domain. Instead, the viral CP contains two domains (S and P) that are constitutively expressed as a single polypeptide[27,28], with the S domain forming the icosahedral capsid shell and the P domain extending from each monomer via a flexible linker at all points of two-fold rotational symmetry within the assembled virion (Fig. S2c, d). While previous cryo-EM studies demonstrated structural homology between polerovirus CPs and tombusvirus S domains (Fig. S2e), structural superposition here shows that the TBSV P domain aligns with the jelly roll domain of each $^N$RTD but lacks the corresponding segments that make up the cap domain (Fig. 1c, d and Fig. S3c). The conserved topologies between both families (Fig. 1b, d) and the intricate distribution of cap domain segments throughout the primary sequence (Fig. S5) suggest that polerovirus structural proteins may be ancestral to those of tombusviruses, with tombusvirus capsids likely evolving via the gradual loss of cap domain elements and truncation of loops L1-L5 rather than through the concerted acquisition of these segments in a manner that would be constrained by the proper folding of both domains.

We also observe a largely unstructured peptide (the 'C peptide') extending from β16 in the TuYV $^N$RTD, which transverses sheet 1 and terminates in a final strand (β17) that packs against β12 in an anti-parallel orientation (Fig. 1a, b, marine). From the electron density, we can define the sequence of this segment unambiguously as the C-terminal portion of the construct (residues 431-459) (Fig. S6a). A disconnected fragment of the C peptide (residues 442-445) is resolved

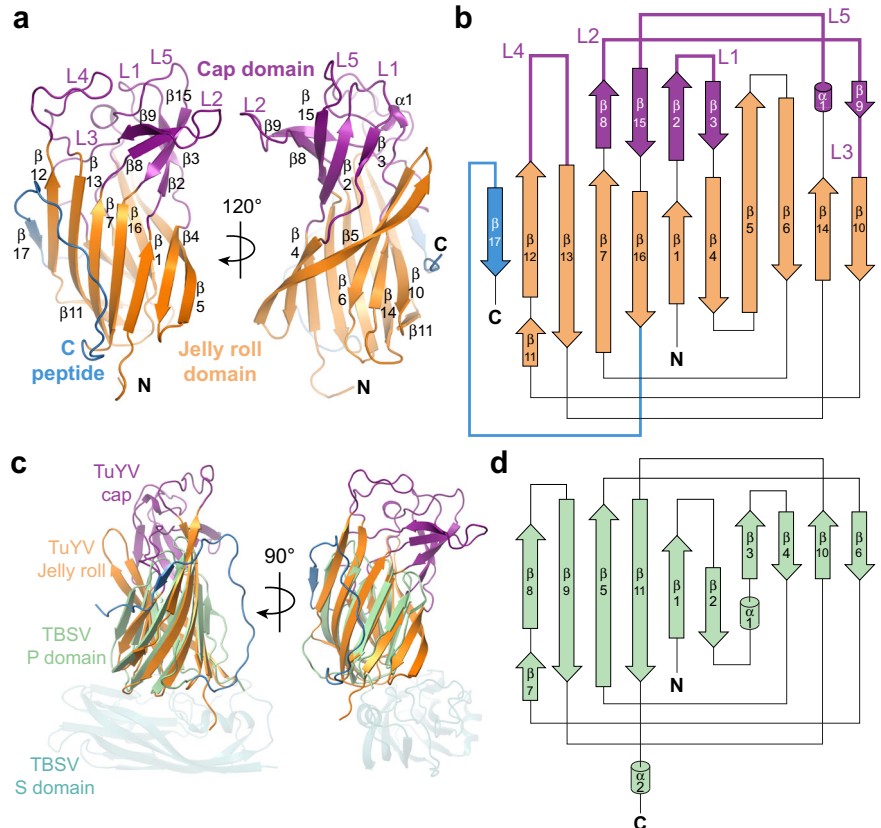

**Fig. 1 | Structure and topology of the TuYV [N]RTD. a, b** Structure (**a**) and topology (**b**) of TuYV [N]RTD with jelly roll domain (orange), cap domain (purple) and C peptide (marine) labeled. Cap domain loops are labeled L1-L5. See Fig. S5 for correspondence between the secondary structure elements and the TuYV sequence. **c** Superposition of TuYV [N]RTD with tomato bushy stunt virus (TBSV) coat protein (PDB: 2TBV; sequence identity: 9% (across the P domain); DALI[26] Z score: 7.0; RMSD: 2.7 Å; P domain, light green; S domain, teal). **d** Topology of TBSV P domain.

in the PLRV structure (Fig. S6b), likely owing to partial proteolytic cleavage and dissociation of the liberated fragment during purification and/or crystallization.

## The C peptide stabilizes the [N]RTD dimer interface

PLRV and TuYV [N]RTDs crystallize as dimers (Fig. S1f, g), consistent with their stoichiometry in solution. Individual monomers superimpose with an overall RMSD ranging from 1.12-1.42 Å across all atoms, with the L2, β4-β5, and β13-β14 loops and portions of the C peptide exhibiting the greatest degree of structural variability (Fig. S3d, e). Within each dimer, [N]RTD monomers are oriented parallel to the dimer symmetry axis with the sheet 1 side of the jelly roll facing inward (Fig. 2a, b and Fig. S7a, b). The dimensions of the TuYV dimer are 63 Å by 62 Å by 42 Å (Fig. S8).

Cap domain loops L4 and L5 form the upper portion of the TuYV dimer interface, with main chain atoms and residues E315, H356, E360, N362, and Y410 (H362, E366, N368, S369, and Y415 in PLRV) making hydrogen bonds in trans (Fig. 2c, d and Fig. S7c, d). Interacting side chains from β12, L2, and β9 provide additional contacts at the edges of the dimer (Fig. 2d and Fig. S7d). The C peptides snake up from the bottom of the TuYV jelly roll, filling the large cavity beneath the cap domains before exiting in opposite directions to wrap around sheet 2 (Fig. S6c). The R440 and R443 side chains anchor an extensive network of stabilizing hydrogen bonds and hydrophobic interactions along the interior of the structure while β17 serves a similar role on the exterior (Fig. 2e and Fig. S6d). Together, the C peptides increase the total buried surface area from 908 Å² to 3615 Å², constituting a major driving force of dimerization. Although we only resolve a partial fragment from one C peptide in the PLRV [N]RTD dimer structure (Figs.

S6b and S7a, b), this piece forms similar stabilizing interactions with both monomers (Figs. S6d and Fig. S7e). Deletion of the C peptide from either [N]RTD expression construct renders the resulting proteins insoluble. ConSurf analysis[29] shows that residues directly contacting the TuYV C peptides are highly conserved across all P/E/L viruses (Figs. S9 and S5), signifying the general importance of these interactions as they are maintained throughout the evolution and adaptation of all three genera. Interestingly, a C-terminal truncation of the CABYV RTP terminating immediately after the C peptide can be efficiently incorporated into mature virions whereas mutants disrupting the anchoring arginines cannot[13] (Fig. S6e). Together these data underscore the critical role the C peptide plays in the proper folding and stability of the [N]RTD dimer.

## Disruption of [N]RTD folding and stability impairs transmission of mutant viruses

Viral mutants have played an integral part in advancing our understanding of P/E/L virus biology, particularly with regard to movement, uptake, and transmission. Our data afford the opportunity to re-examine these perturbations in a structural context and, consequently, reinterpret the associated phenotypes. Systematic mutation of conserved residues throughout the PLRV [N]RTD, including a series of triplet residue deletions (Fig. S10), previously yielded some mutants where the RTP was not incorporated into the assembled virion and other mutants that were incorporated but were not aphid transmissible[14] (Table S2). Many of the mutated side chains are buried and form stabilizing hydrogen bonding and hydrophobic interactions (Fig. S10a, b), suggesting that the triplet deletions interfere with the structural integrity and folding of the [N]RTD dimer. To test this hypothesis, we

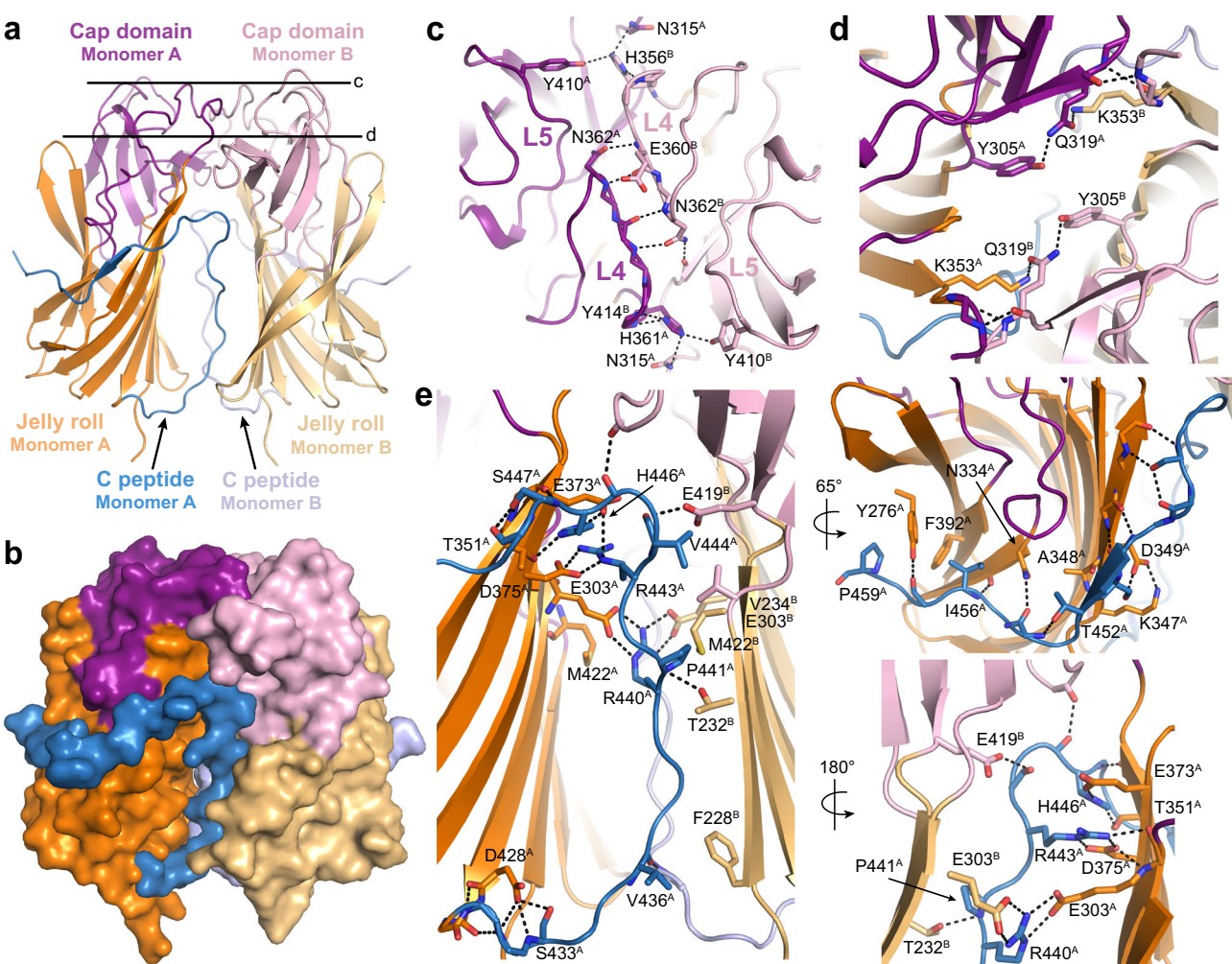

**Fig. 2 | Architecture of the TuYV $^{N}$RTD dimer. a, b** Cartoon (**a**) and surface (**b**) representations of the TuYV $^{N}$RTD dimer. Individual structural segments are labeled in each monomer and colored as follows: Jelly roll domains, orange and light orange; cap domains, purple and light pink; C peptides, marine and light blue. **c, d** Slice sections through the dimer at the levels indicated by the solid lines in (**a**) highlighting stabilizing interactions at the dimer interface. Dashed black lines denote hydrogen bonds. Key residues are labeled with a superscript (A or B) to indicate from which monomer they originate. Secondary structure elements (see Fig. 1) are labeled where applicable. **e** C peptide interactions. Residues contributing hydrogen bonding (dashed black lines) and hydrophobic contacts are labeled.

introduced a subset of the triple deletion and triple alanine mutations from previous studies into our PLRV $^{N}$RTD construct (unincorporated: $^{241}$PML$^{243}$, $^{409}$YNY$^{411}$; incorporated: $^{233}$RFI$^{235}$, $^{268}$EDE$^{270}$, $^{315}$SST$^{317}$) and examined the solubility of the resulting proteins. Western blot analysis was used to detect the $^{N}$RTD in the soluble (supernatant) and insoluble (pellet) fractions following recombinant expression in *E. coli*. The two non-incorporated mutants (PML and YNY) were insoluble in this context, as neither was detected in the supernatant (Fig. S10d, e). The incorporated EDE mutants were similarly insoluble while the SST mutants were partially soluble. Only the ΔRFI mutant remained soluble though a triple alanine mutant at this same position was not. These observations argue that the in vivo defects associated with mutant viruses arise from disruption of the proper folding and/or stability of the $^{N}$RTD, which is critical for aphid transmission.

Point mutations in the RFI, EDE, and YNY motifs produced similar transmission defects in TuYV[30] (Table S2). An alanine substitution at R227 (R233 in PLRV RFI triplet) reduced transmission while a double alanine mutant at E262 and D263 (E268 and D269 in PLRV EDE triplet) was only transmissible after microinjection, meaning the mutant virus was unable to traverse the aphid gut. As in PLRV, these side chains participate in a network of hydrogen bonds that stretches between β1, β4 and β6 and buttresses sheets 1 and 2 at the bottom of the jellyroll domain (Fig. S10b, c). Interestingly, a compensatory second site

mutation converting proline 235 to a leucine restores both the infectivity and transmission of the R227A and E262A/E263A TuYV mutants[30] (Table S2). P235 sits in an unstructured segment between β1 in the jelly roll domain and β2 in the cap domain, facing into a hydrophobic pocket lined with F259, I288, I421, and I424 (Fig. S10c). We speculate that a leucine substitution at this position would alter the overall secondary structure and/or strengthen the existing hydrophobic interactions to keep the cap domain in place, ultimately overcoming any instability in the distal portions of the fold.

The K403A/Y404D (Y409 in PLRV YNY triplet) double mutant also showed reduced TuYV transmission, which could be improved by microinjection[30]. These conserved side chains lie on the edge of the cap domain in both structures where K403 helps anchor L4 and Y404 lines the wall of a cavity on the surface (Figs. S9d, S10c). Revertant mutations switching Y404D back to a tyrosine or structurally similar phenylalanine rescue the transmission defect, suggesting a pi-cation interaction with Q399 is important for the stabilization of this region of the protein[31].

## $^{N}$RTD architecture does not limit stoichiometry in the context of the mature virion

Stochastic ribosomal readthrough of the CP stop codon sub-stoichiometrically limits the amount of RTD that is translated, and,

hence, present for incorporation into mature, infectious virions[32]. Why the stop codon has been evolutionarily maintained in the virus genome despite the critical role for the $^N$RTD in aphid transmission is unknown. Leveraging the observed homology with tombusvirus S and P domains (Fig. 1, Figs. S2, and S3c), we modelled the organization of the $^N$RTD on the capsid surface to identify possible restraints on virion assembly (Fig. 3). Tombusvirus P domains are constitutively translated and tethered to each S domain via an unstructured linker (Fig. 3a). When assembled, the P domains occupy every two-fold symmetry axis in the $T = 3$ icosahedral capsid (Fig. 3b and Fig. S2c). We anticipate that intact RTPs encoded by P/E/L viruses will follow the same architectural design but with the added constraint of head-to-head $^N$RTD dimerization imposed. A composite model combining the TuYV $^N$RTD and CP (PDB: 6RTK) coordinates suggests a similar overall connectivity (Fig. 3c), with the $^N$RTD dimer situated about the two-fold symmetry axis but rotated approximately 15° relative to the position of the TBSV P domains (Fig. 3d–f). This organization confirms previous predictions that special interactions stabilize the association of icosahedral asymmetric units (Fig. 3f, black triangles) across the two-fold axis of symmetry and may contribute to the overall pathway of virion assembly[21]. Importantly, we note no steric clashing if this RTP model is placed at each position in the $T = 3$ icosahedral asymmetric unit (Fig. 3g). This implies that the $^N$RTD could feasibly occupy every two-fold position in a polerovirus capsid (yielding a total of 90 $^N$RTD dimers) and that the architecture of the $^N$RTD itself does not intrinsically limit its stoichiometry. The proximity of this arrangement, however, might be problematic in that it could promote aggregation and/ or collision between the disordered C-terminal region of the RTD in neighbouring subunits, ultimately destabilizing the structure or masking segments of the RTD that may interact with aphid receptors. We speculate that the leaky CP stop codon is therefore preserved to ensure a low concentration of this bulky C-terminal extension on the virion surface. $^N$RTD dimerization might also impose kinetic constraints that further limit RTP incorporation into the capsid if the timescale of folding is slower than the rate of CP assembly.

## $^N$RTD can function as an inhibitor of viral transmission

A soluble version of the tomato spotted wilt virus (Genus: *Orthotospovirus*; Family: *Tospoviridae*) membrane surface glycoprotein $G_N$ was shown in feeding experiments to inhibit viral transmission by its insect vector, the western flower thrips, *Frankliniella occidentalis*[33,34]. Given the important role of the $^N$RTD in aphid transmission, we asked whether the purified PLRV $^N$RTD dimer could similarly compete with the mature virus for binding to aphid tissues and subsequently hinder its uptake and transmission by its primary vector, the green peach aphid, *Myzus persicae*. To test this, we first exposed *M. persicae* to the purified $^N$RTD in an artificial diet feeding system prior to PLRV acquisition and then monitored the subsequent transmission to healthy plants (Fig. S11). Transmission of PLRV to potato was significantly decreased under these conditions as compared to the no protein control (Fig. 4a, Tables S3, S4, $P = 0.046$, Wald $z$ test), despite the fact that oral delivery of the same concentration of the purified bovine serum albumin (BSA) control significantly increased transmission ($P = 0.035$ compared to the no protein control, Wald $z$ test; $P < 0.0001$ compared to $^N$RTD, Wald $\chi^2$ test), a well-described phenotype in the literature observed for proteins unrelated to aphids or virus transmission, including BSA, casein, lysozyme and cytochrome C[35].

Next, we tested whether transient expression of the $^N$RTD *in planta* using *Agrobacterium tumefaciens* would also block virus transmission (Fig. S12). Expression tests and western blot analysis showed that the PLRV $^N$RTD requires a small protein tag to facilitate folding *in planta* (Fig. S12a–d), an unsurprising finding as native $^N$RTD would be fused to the CP on its N-terminus and the $^C$RTD on its C-terminus. *M. persicae* were allowed to feed on *Nicotiana benthamiana* leaves transiently expressing YFP-tagged PLRV $^N$RTD before testing their ability to

transmit virus (Fig. S12e). We tested both N-terminal and C-terminal YFP tags and found that aphids pre-exposed to the YFP-$^N$RTD by this delivery method also had a decreased ability to transmit virus (Fig. 4b, Tables S5 and S6, $P = 0.011$ compared to uninfiltrated control, Wald $z$ test).

Considering the promising results of transient *in planta* expression, we generated transgenic potato plants constitutively expressing YFP-$^N$RTD under the control of the cauliflower mosaic virus (CaMV) 35 S constitutive promoter. Expression of YFP-$^N$RTD in these plants was confirmed via western blot analysis, RT-PCR, and fluorescence confocal microscopy (Fig. S13a–c). YFP signal was observed along the cell periphery and in the nucleus of the YFP-$^N$RTD transgenics but not in the empty vector controls (Fig. S13c). *M. persicae* were exposed to transgenic leaves for 48 hours as in previous experiments and then PLRV titer in aphids was quantified by droplet digital PCR (Fig. S13d) and their ability to transmit PLRV was assessed (Fig. S13e). Aphids exposed to YFP-$^N$RTD acquired significant fewer copies of PLRV (Fig. 4c, $P = 0.038$, unpaired one-sided Student's $t$ test) and had a reduced ability to transmit PLRV (Fig. 4d, Tables S7 and S8, $P = 0.044$, Wald $z$ test). These results are consistent with the outcomes from the other delivery strategies and show that the reduction in PLRV transmission is likely due to the reduced ability of aphids to acquire virions across the midgut barrier.

A meta-analysis of our data found that pre-treatment of aphids with the PLRV $^N$RTD significantly reduced the chances of a plant from becoming infected by nearly half (risk ratio of 0.55) with the 95% confidence interval ranging from a 25% reduction in infection (risk ratio of 0.75) to a 60% reduction (risk ratio of 0.40, Fig. S14) regardless of delivery route. There was remarkably low heterogeneity between experiments and even among delivery methods ($I^2 = 0\%$, $\tau^2 = 0$, $P = 0.42$, Cochran's $Q$ test), indicating that this effect is highly reproducible even when using different delivery strategies. These results argue that the isolated $^N$RTD can function as an inhibitor of viral transmission.

## Cap domain mutants are lethal to aphids

Our modelling suggests that the $^N$RTD protrudes from the virion surface with the cap domain directed outward (Figs. 2a, 3d, f), poised to make direct contact with aphid receptors that contribute to viral transmission[19]. We reasoned that mutating surface-exposed side chains within this domain would disrupt critical interactions needed for viral uptake and thus could impair the ability of the $^N$RTD to function as an inhibitor in our transmission assays. To test this hypothesis, we introduced a series of alanine substitutions into the PLRV $^N$RTD, including point mutations at non-conserved residues H321, E366, H371, E374, and a "cluster" mutant containing three mutations at N368, C370, and Y411, which form a highly conserved pocket (Fig. 5a and Fig. S15). These positions were chosen as they do not interfere with $^N$RTD dimerization and folding. $^N$RTD mutants were purified and delivered to aphids via artificial diet feeding prior to PLRV acquisition and then viral transmission to healthy plants was measured by ELISA (Fig. S11a). As predicted, the H321A mutation interfered with the inhibitor function of the $^N$RTD and did not significantly decrease PLRV transmission compared to the no protein control ($P = 0.817$; unpaired two-tailed Student's $t$ test) (Fig. 4a). This behavior was distinct from the increase in transmission observed with the BSA control (Fig. 4a, compared to BSA, $P = 0.140$, unpaired two-tailed Student's $t$ test), providing further support that the observed phenotypes are specific for the $^N$RTD and not a general consequence of ingesting protein.

While attempting to assay the other cap domain mutants, we noted that aphids died at a significant rate. To quantify this mortality, aphids were fed the $^N$RTD mutants via an artificial diet and then moved to either PLRV-infected or uninfected leaves for 24 h after which the numbers of live and dead insects were counted (Fig. S11b, Table S9).

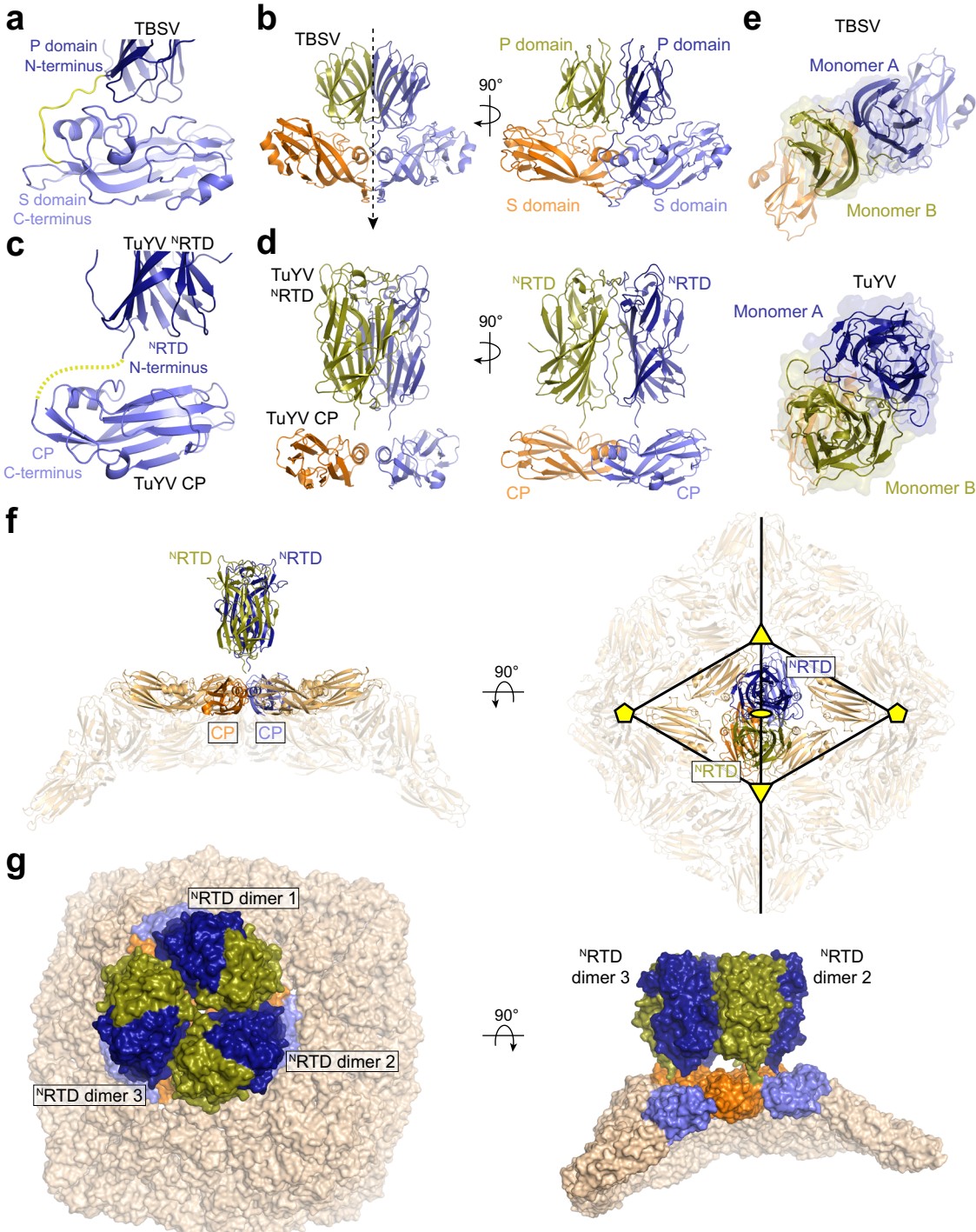

**Fig. 3 | $^N$RTD architecture does not limit stoichiometry in the context of the mature virion. a** Domain connectivity in TBSV capsid proteins. Unstructured linker that connects the C-terminus of the S domain (light blue) to the N-terminus of the P domain (dark blue) highlighted in yellow. **b** Arrangement of TBSV capsid proteins at the two-fold symmetry axis (dashed arrow) in the assembled virion (see Fig. S2) shown in two orientations. S and P domains associated with individual monomers are colored orange and olive (monomer A) and slate and dark blue (monomer B). **c** Predicted connectivity in TuYV capsid proteins based on structural modeling. Dashed yellow line denotes the predicted trajectory linking the C-terminus of the TuYV CP (light blue, PDB: 6RTK) to the N-terminus of the TuYV $^N$RTD (dark blue). **d** Composite model of the polerovirus RTP built from the crystallized TuYV $^N$RTD dimer and CP monomers taken from the cryo-EM reconstruction of modified TuYV

virion devoid of the readthrough domain (PDB: 6RTK). RTP dimer is organized around two-fold symmetry axis analogous to the arrangement in (**b**) (see Fig. S2). **e** View of subunit associations in (**b**, **d**) looking down the two-fold axis of symmetry in the direction of the dashed arrow in (**b**). **f** Side (left) and top down (right) views of TuYV RTP modeled at the two-fold symmetry axis of the icosahedral virion. Two-, three-, and five-fold symmetry axes are marked with a yellow ellipse, yellow triangles, and yellow pentagons, respectively. RTP is colored as in (**d**) with the rest of the capsid subunits colored wheat. **g** Model illustrating the feasible positioning of $^N$RTD dimers (olive and dark blue) around icosahedral asymmetric unit of TuYV VLP assuming the structural organization of the RTP in (**d**). Associated CP monomers are colored orange and slate with the rest of the capsid surface colored wheat.

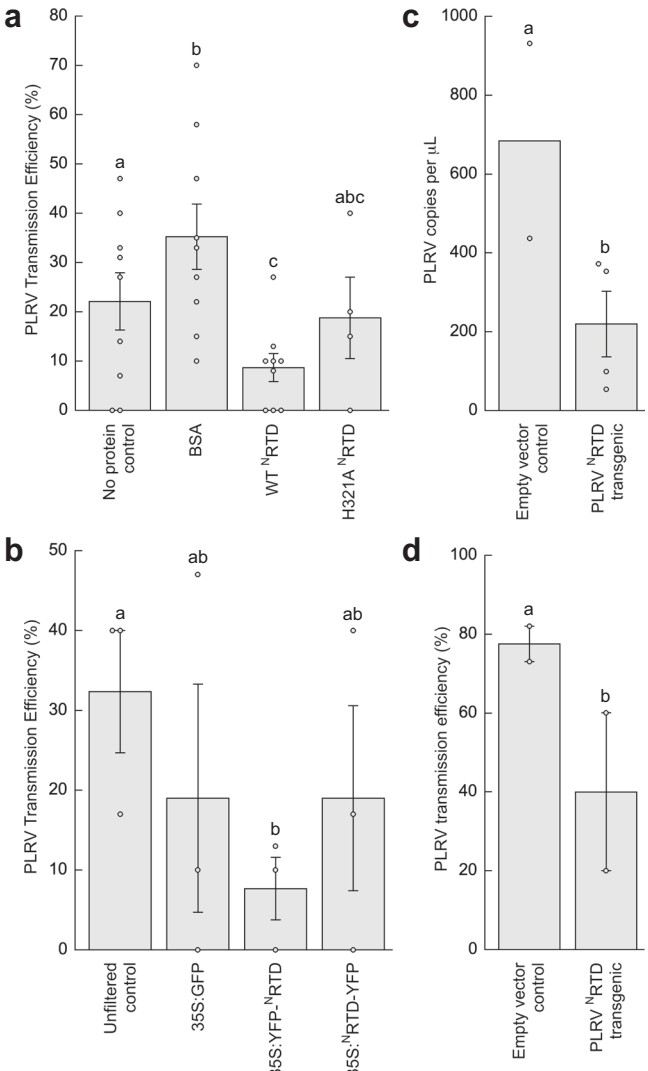

**Fig. 4 | The $^N$RTD can function as an inhibitor of viral transmission. a** The PLRV transmission efficiency of *Myzus persicae* is significantly different after feeding on various artificial diet treatments (see Fig. S11a): no protein control ($n = 101$ inoculated plants), BSA ($n = 116$), purified WT PLRV $^N$RTD ($n = 124$), or PLRV $^N$RTD point mutant H321A ($n = 43$). **b** PLRV transmission efficiency of *M. persicae* aphids is reduced after transient *in planta* delivery of the PLRV $^N$RTD ($n = 37$ inoculated plants for all treatments, see Fig. S12). **c** Aphid acquisition of PLRV is reduced after exposure to the PLRV $^N$RTD transgenic ($n = 4$ pools of 5 insects) compared to empty vector control ($n = 2$) potato plants as quantified via droplet digital PCR (see Fig. S13). Mean ± one standard error for all replicates (dots) is shown. Letters above each bar represent significantly different treatments ($P = 0.038$) by an unpaired one-sided Student's $t$ test. d, PLRV transmission efficiency of *M. persicae* aphids is reduced after exposure to the PLRV $^N$RTD transgenic ($n = 60$ inoculated plants) compared to empty vector control ($n = 26$) potato plants ($P = 0.044$, see Fig. S13). For panels (**a**), (**b**), (**d**), the mean ± one standard error for all independent repeats of the experiment (dots) is shown. Each inoculated plant was considered a biological replicate. Letters above each bar represent significantly different treatments ($P < 0.05$) by logistic regression analysis using a one-sided Wald $χ^2$ test of model coefficients. Full models, model diagnostics, test statistics, degrees of freedom, and exact $P$-values are reported in Supplementary Tables S4 (**a**), S6 (**b**), and S8 (**d**).

E366A, H371A, E374A, and the cluster mutant all caused significant mortality (Fig. 5b, Table S10, $P < 0.001$ for all four mutants, respectively, as compared to no protein control), with E374A and the cluster mutant having the strongest effects. Aphids died even when transferred from the $^N$RTD mutant laden diet treatments to uninfected leaves (Fig. 5b), showing that the observed mortality was linked

exclusively to the purified $^N$RTD variants and independent of infectious virus.

The H321A mutant did not cause significant aphid death when compared to no protein and BSA controls ($P > 0.44$, Fig. 5b). The fact that this mutation localizes to a different region of the cap domain (Fig. 5a) and has different effects with respect to viral transmission and aphid mortality hints that the $^N$RTD interacts with the aphid gut in a structure-specific manner. Differential binding of the WT $^N$RTD and the mutants to various aphid proteins is one hypothesis that may explain both the transmission assay and aphid mortality data. Regardless of the mechanism, the ability of these mutants to kill aphids means they can be deployed as biopesticides, either through transgenic plant delivery as described above for the wild-type $^N$RTD or via some other delivery strategy.

## Mechanistic and translational implications of our findings

Our work here defines the basic structural organization of polerovirus $^N$RTDs and provides a model for how the RTP is incorporated into the mature virion. Polerovirus $^N$RTDs adopt an interdigitated two-domain architecture and form dimers that are stabilized by a C-terminal peptide. The requirement for dimerization suggests that the $^N$RTD is situated on the two-fold symmetry axis of the icosahedral viral capsid. Our modeling, however, suggests that this arrangement does not intrinsically limit the stoichiometry of the RTD within the mature virus, which instead may be a consequence of potential $^C$RTD aggregation and/or kinetic constraints imposed by $^N$RTD folding and dimerization. Our data also rationalize the effects of various RTD mutants that have been reported over the last several decades[16,18,36–38]. We now can attribute the observed transmission defects associated with different deletions and truncations to the production of insoluble forms of the $^N$RTD.

Previous studies have used the leaky CP stop codon as the starting point for the readthrough domain. Our work here, however, establishes the $^N$RTD as a defined structural unit that begins downstream of this juncture (at residue 230 in PLRV and 224 in TuYV) and is conserved across all P/E/L viruses (Fig. S5). The intervening sequence that lies between the leaky stop and the beginning of our $^N$RTD crystallographic models is variable and predicted to be unstructured when analyzed by a variety of modelling algorithms (e.g., Phyre, RaptorX, ITASSer, JPRED, GlobPlot, etc.). Given these observations, we propose that the $^N$RTD be defined based on the boundaries elucidated here and that the intervening sequence immediately following the leaky CP stop codon be henceforth referred to as a "variable linker" (Fig. S5). We anticipate these definitions will be more consistent for the field moving forward, especially as we begin to explore the interactions and applications of $^N$RTD constructs deriving from other P/E/L viruses.

Structural comparisons revealed an unexpected evolutionary connection to tombusviruses, intimating that poleroviruses more closely resemble a common ancestor and that loss of the cap domain decouples a virus from its obligate vector and coincides with the ability to be transmitted in other ways. In 2021, the International Committee on Taxonomy of Viruses abolished the previous designation of P/E/L viruses as a single family, *Luteoviridae*, recategorizing luteoviruses as *Tombusviridae* and poleroviruses and enamoviruses as *Solemoviridae* based solely on differences in their respective RNA-dependent RNA polymerases[39]. Our data argue that P/E/L viruses are in fact structurally and mechanistically distinct from other members of these families and should therefore be treated as a separate group when considering the assembly and organization of the capsid and the mode of transmission.

There are presently no treatments to cure plants of polerovirus infections and current methods to breed viral disease-resistant crops or to control aphid vector populations have proven ineffectual to manage these viruses in the field. Our functional experiments demonstrate that the purified PLRV $^N$RTD can act as an inhibitor to

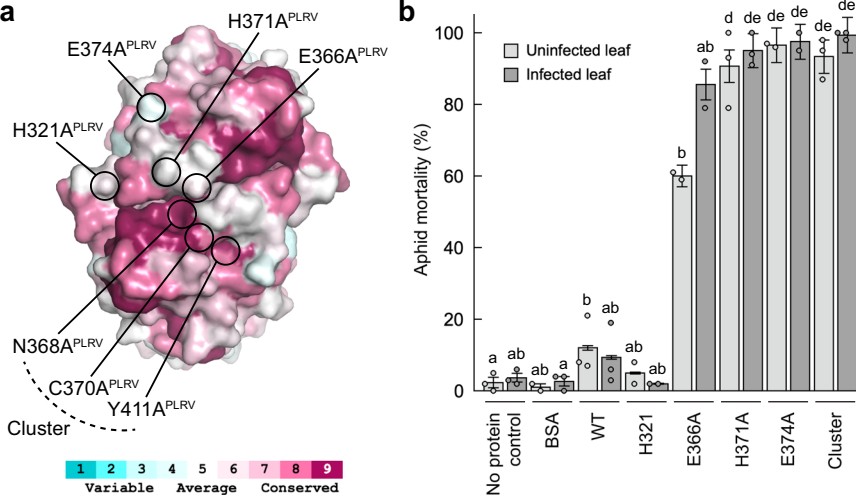

**Fig. 5 | Cap domain mutants are lethal to aphids. a** Top view of TuYV [N]RTD dimer illustrating positions of PLRV cap domain mutants. Coloring reflects sequence conservation among polero-, enamo-, and luteoviruses (legend below, see also Figs. S5 and S9) and was generated using the ConSurf Server[29]. See Fig. S15 for further representation of where the mutants localize within the PLRV [N]RTD structure. **b** *Myzus persicae* mortality after feeding on 0.1 mg/mL of BSA ($n = 232$ insects), purified WT PLRV [N]RTD ($n = 247$), PLRV [N]RTD point mutants H321A ($n = 183$), E366A ($n = 118$), H371A ($n = 264$), E374A ($n = 161$), a PLRV [N]RTD cluster mutant (containing mutations N368A, C370A, and Y411A; $n = 256$) or no protein controls ($n = 244$) for 48 h and then moved to an uninfected (light gray) or PLRV-infected (dark gray) detached hairy nightshade leaf (See Fig. S11b). Mean ± one standard error for all independent repeats of the experiment (dots) is shown. Each individual aphid was considered a biological replicate. Different letters represent significantly different treatments ($P < 0.05$) by quasibinomial regression analysis using a one-sided Wald $\chi^2$ test of model coefficients. Full models, model diagnostics, test statistics, degrees of freedom, and exact $P$-values are reported in Supplementary Table S10.

reduce its vector transmission, not only in the context of artificial diet feeding in the laboratory but also when but delivered transgenically *in planta* in the greenhouse. This provides a biologically based strategy for controlling these insect-transmitted viruses that can be deployed for widespread field application pending regulatory approval. The high degree of sequence conservation across the jelly roll and cap domains, particularly in regions contacting the C peptide, implies that enamovirus and luteovirus [N]RTDs will follow the same structural blueprint and that this inhibitor-based approach could be extended to mitigate these agricultural pathogens as well. Moreover, we show that certain point mutations in the cap domain are lethal to the insect vector, indicating we have discovered a potent biopesticide with many possible permutations and the use of transgenic plants as an already proven delivery strategy. The deployment of a genetically encoded insecticidal protein that also blocks virus transmission in crop plants lies at the forefront of agricultural technology[40,41] and may one day eliminate the need for environmentally harmful and costly pesticide applications. Our work here represents a major step forward toward achieving this end.

Taken together, our findings advance our fundamental understanding of plant virology and vector biology and impart new tools that can be used to thwart both vector-borne phytoviruses and an economically damaging group of insects.

## Methods

### Cloning, expression, and purification of polerovirus [N]RTD constructs

DNA encoding the N-terminal portion of the potato leaf roll virus readthrough domain (PLRV [N]RTD; residues 230-458) and the N-terminal portion of the readthrough domain of turnip yellows virus (TuYV [N]RTD; residues 224-459) were amplified by PCR and cloned into pCAV4, a modified T7 expression vector that introduced an N-terminal 6xHis-NusA tag followed by a HRV3C protease site upstream of the inserted sequence. Constructs were transformed into BL21(DE3) cells, grown at 37 °C in Terrific Broth to an $OD_{600}$ of 0.7-0.9, and then induced with 0.3 mM IPTG overnight at 19 °C. Cells were pelleted, washed with nickel loading buffer (20 mM HEPES pH 7.5, 500 mM NaCl, 30 mM imidazole, 5% glycerol (v:v), and 5 mM

β-mercaptoethanol), and pelleted a second time. Pellets were typically frozen in liquid nitrogen and stored at −80 °C.

Selenomethionine labeled protein was expressed in T7 Express Crystal *E. coli* (BL21 derivative, New England Biolabs) according to the manufacturer's recommended protocol. Briefly, a 2 L culture of minimal media (1X M9 salts, 0.4% glucose, 0.1 mM $CaCl_2$, 2 mM $MgSO_4$, 0.0002% ferric ammonium citrate) supplemented with methionine (50 μg/mL) was grown to an $OD_{600}$ of 0.7. The media was then removed following centrifugation and replaced with minimal media lacking methionine. After three hours of incubation at 37 °C, the media was supplemented with 50 μg/mL L-selenomethionine (Sigma) and induced overnight at 19 °C with 0.3 mM IPTG (Millipore).

Thawed 500 ml pellets of native and SeMet [N]RTD constructs were resuspended in 30–40 ml of nickel loading buffer supplemented with DNAse (0.25 mg/ml, Roche), 1 mM $MgCl_2$, 10 mM PMSF, and a Roche complete protease inhibitor cocktail tablet. Lysozyme was added to a concentration of 1 mg/ml and the mixture was incubated for 10 minutes rocking at 4 °C. Cells were disrupted by sonication and the lysate was cleared via centrifugation at 13,000 rpm (19,685 × *g*) for 30 min at 4 °C. The supernatant was filtered, loaded onto a 5 ml HiTrap chelating column charged with $NiSO_4$, and then washed with nickel loading buffer. Native and SeMet [N]RTD were eluted by an imidazole gradient from 30 mM to 500 mM. Peak fractions were pooled and HRV3C protease was added to a final concentration of 0.5 units/mg of fusion to remove the 6xHis-NusA. Pooled fractions and protease were dialyzed overnight at 4 °C into Q loading buffer (20 mM TRIS pH 8.0, 50 mM NaCl, 1 mM EDTA, 5% glycerol (v:v), and 1 mM DTT). The dialyzed sampled was applied to 5 ml HiTrap Q HP column equilibrated with Q loading buffer, washed in the same buffer, and eluted with a NaCl gradient from 50 mM to 500 mM. Peak fractions were pooled, concentrated, and further purified by size exclusion chromatography (SEC) using a Superdex 75 10/300 column. Native and SeMet PLRV [N]RTD were exchanged into a final buffer of 20 mM HEPES pH 7.5, 150 mM KCl, and 1 mM DTT during SEC and concentrated to 20–30 mg/ml. Native TuYV [N]RTD was purified in the same manner.

All PLRV [N]RTD mutants were generated by QuikChange mutagenesis (Agilent Technologies) and confirmed by sequencing. Cap

domain mutants for aphid feeding experiments (H321A, H366A, H371A, H374A, N368A/C370A/Y411A) were expressed and purified as described above for the native WT PLRV [N]RTD. Triple mutant [N]RTD constructs (Fig. S9) were further cloned into pET15b for expression tests and western blot analysis (see below).

## Size exclusion chromatography coupled to multiangle light scattering

The oligomeric state of PLRV and TuYV [N]RTD was determined by size-exclusion chromatography coupled to multi-angle light scattering (SEC-MALS). The [N]RTD proteins were loaded at 4 mg/mL onto a Superdex 200 10/300 Increase column (GE) in size exclusion buffer (20 mM HEPES pH 7.5, 150 mM KCl, and 1 mM DTT) at a flow rate of 0.5 ml/min. Eluent from the sizing column flowed directly to a static 18-angle light scattering detector (DAWN HELEOS-II) and a refractive index detector (Optilan T-rEX) (Wyatt Technology) with data collected every second. Molar mass was determined using the ASTRA VI software. Monomeric BSA (Sigma) was used for normalization of light scattering.

## Crystallization, X-ray data collection, and structure determination

SeMet PLRV [N]RTD at 6 mg/ml was crystallized by sitting drop vapor diffusion at 20 °C in 0.1 M Tris-HCl pH 8.5, 0.25 M Ammonium sulfate, 14% polyethylene glycol 8000, 3% ethylene glycol, and 5 mM DTT. Crystals were of the space group $P2_122_1$ with unit cell dimensions $a = 63.229$ Å, $b = 65.147$ Å, $c = 109.681$ Å and $\alpha = 90°$, $\beta = 90°$, $\gamma = 90°$ and contained a dimer in the asymmetric unit. Samples were cryo-protected with Parabar 10312 prior to freezing in liquid nitrogen. Crystals were screened and optimized remotely using NE-CAT beamlines at the Advanced Photon Source. Single-wavelength anomalous diffraction (SAD) data[24] were collected remotely on the 24-ID-E NE-CAT beamline at the selenium edge ($\lambda = 0.9792$ Å) at 100 K to a resolution of 2.21 Å (Table S1). Data were integrated and scaled via the NE-CAT RAPD pipeline, using XDS[42] and AIMLESS[43], respectively. Experimental phasing was carried out remotely from a beach in Malta using the RAPD pipeline and a surprisingly strong hotel Wi-Fi signal. A total of 13 selenium sites were found by SHELX[44] and used for initial phasing with a figure of merit of 0.29. Density modification and initial model building was carried out using the Autobuild routines of the PHENIX package[45]. Further model building and refinement was carried out manually in COOT[46] and PHENIX[45]. The final model was refined with $R_{work}/R_{free}$ values of 0.20/0.25 and contained residues 230-436 in monomer A, residues 232-435 in monomer B, a short peptide (C peptide) consisting of residues 442-458, one sulfate ion, and 16 water molecules. The model contains 96.2% Ramachandran favored resides, 3.6% allowed, and 0.2% outliers.

Native TuYV [N]RTD at 12 mg/ml was crystallized by sitting drop vapor diffusion at 20 °C in 0.1 M HEPES pH 7.7, 1.4 M AmSO₄, 0.1 M NaCl. Crystals were of the space group $P2_12_12_1$ with unit cell dimensions $a = 46.4$ Å, $b = 74.9$ Å, $c = 130.8$ Å and $\alpha = 90°$, $\beta = 90°$, $\gamma = 90°$ and contained a dimer in the asymmetric unit. Samples were cryoprotected by transferring the crystal directly to Parabar 10312 (Hampton Research) prior to freezing in liquid nitrogen. Diffraction data were collected remotely on the NE-CAT 24-ID-C beamline at the Advanced Photon Source at the selenium edge ($\lambda = 0.9791$ Å) at 100 K to a resolution of 1.53 Å (Table S1). Data was processed as above. Molecular replacement was carried out with Phaser-MR in the PHENIX package[45] using residues 230-436 of the PLRV N-RTD crystal structure (PDB: 7ULO). PHENIX Autobuild[45] was used for the initial model building; manual model building and refinements were carried out in COOT[46] and PHENIX[45]. The final model was refined with $R_{work}/R_{free}$ values of 0.17/0.20 (Table S1). Both monomers were modelled with residues 225-460, along with 587 water molecules. The final model has 98.5% Ramachandran favored residues and 1.5% allowed.

Structural superpositions and viral capsid modeling were carried out in Chimera[47] and surface electrostatics were calculated using APBS[48]. All structural models were rendered using Pymol (Schrodinger). Visual mapping of conserved residues was carried out using the ConSurf server[29].

## Expression tests and western blot analysis of [N]RTD triple mutants

To avoid any confounding effects from the NusA solubility tag, wild-type and triple alanine or deletion mutant PLRV [N]RTD constructs (Fig. S9) were cloned into the pET15b expression plasmid, introducing an N-terminal 6xHis tag. Constructs were transformed into BL21(DE3) cells and grown in 50 mL of LB broth at 37 °C to an OD₆₀₀ of 0.6. Protein expression was induced with 0.3 mM IPTG overnight at 19 °C. Cells were harvested and resuspended in 30 mL of lysis buffer (20 mM HEPES pH 7.5, 150 mM NaCl, 1 mM DTT). Following a 10-minute incubation with 0.1 mg/mL lysozyme at 4 °C, cells were lysed via sonication. Insoluble material was pelleted by centrifugation at 13,000 rpm (19,685 × g) for 30 min at 4 °C. The soluble fraction was collected and the insoluble pellet was resuspended in lysis buffer. Samples from the soluble and insoluble protein fractions were combined with Laemmli buffer (BioRad) and separated by electrophoresis through a 12% SDS polyacrylamide gel. Proteins were wet transferred to a nitrocellulose membrane and blocked with 5% milk in Tris-buffered saline (TBST, 25 mM Tris, 150 mM NaCl, 0.05% Tween-20) for two hours at 22 °C, followed by incubation overnight at 4 °C with 1:5000 dilution of a polyclonal rabbit antibody raised against the purified PLRV [N]RTD, diluted in TBST. After washing with TBST, blots were incubated with 1:3000 HRP-conjugated goat anti-rabbit IgG (Promega W4011) for 1 h at 22 °C. After washing with TBST, blots were developed with the ECL Prime Western Blot Detection Reagent (New England Biolabs) and visualized with a ChemiDoc (BioRad).

The [N]RTD polyclonal antibody was generated by Cocalico Biologicals by injecting a rabbit with crystallography-grade purified [N]RTD protein. The antibody was cross absorbed against *E. coli*, *N. benthamiana*, and *M. persicae* protein homogenates to remove nonspecific antibodies. The antibody was purified via sodium sulfate precipitation and specificity was verified by western blot analysis against *E. coli*, *N. benthamiana* and *M. persicae* protein homogenates, using the purified [N]RTD as a positive control.

## Artificial diet delivery of the PLRV [N]RTD

The potato leafroll virus sequence used for recombinant protein expression is from a cDNA infectious clone developed by Franco-Lara, et al.[49]. This infectious clone was used to inoculate hairy nightshade (*Solanum sarrachoides*, HNS)[50], for use as a source of inoculum for all virus experiments[50]. The parthenogenetic clone of *Myzus persicae* Sulz used in these experiments, originally collected from New York state, was maintained on *Physalis floridana*. Fourth instar and adult insects were used for all experiments unless otherwise specified.

Artificial diet delivery of WT PLRV [N]RTD, H321A, and the control protein bovine serum albumin (BSA, BioRad), was achieved by diluting the proteins to 0.1 mg/mL or 1 mg/mL in an artificial sucrose diet for *M. persicae* supplemented with amino acids[51]. Diet with no added protein was used as a control. After starving for 1–2 h, *M. persicae* aphids were placed in dishes that were sealed by stretching Parafilm over the top and sandwiching the diet beneath a second piece of Parafilm. Aphids were allowed 48 h of feeding on the diet treatments.

Immediately following artificial diet feeding, aphids were transferred to detached PLRV-infected HNS leaves for a 48-hour acquisition access period (AAP, eight artificial diet feeding experiment only) or 24-hour AAP (all other experiments). After the AAP, 3 aphids per plant (the first two oral delivery experiments) or 5 aphids per plant (all other experiments) were transferred to uninfected potato seedlings (*Solanum tuberosum* cv. Red Maria, 6–15 plants/treatment) for a 72-hour

 

inoculation access period (IAP). Potato seedlings were treated with pymetrozine (Endeavor) and bifenthrin (Talstar P) after the IAP to remove aphids. Systemic PLRV infection was accessed three weeks later by double antibody sandwich enzyme-linked immunosorbent assay (DAS-ELISA) using a 1:200 dilution of a polyclonal antibody generated towards purified PLRV (Agdia SRA30002). Each inoculated plant represents a replicate. Exact sample size for each experiment is given in Tables S3 and S5. The artificial diet experiment was repeated nine times independently.

### Transient in planta delivery of the PLRV $^N$RTD

For *in planta* delivery, transient expression constructs were generated by cloning the PLRV $^N$RTD sequence into pEarlyGate binary expression vectors pEarleyGate101 and pEarleyGate104[52], creating untagged and YFP-tagged versions of the PLRV $^N$RTD, with YFP adhered to the N- or C-terminus. Expression and solubility of these constructs was tested *in planta* via agroinfiltration into *N. benthamiana*. Three leaves per plants, 3 plants per construct were infiltrated, and leaf discs taken at 2-, 3-, and 4-days post inoculation (dpi) for subsequent protein extraction and western blot analysis. Protein was extracted by cryogenic grinding of leaf discs for 6 min at 25 Hz in a Mixer Mill 440 (Retsch) followed by resuspension in extraction buffer (0.1 M Trist pH 8.0, 150 mM NaCl, 20 mM HEPES pH 7.0). Protein extracts were combined with Laemmli buffer (BioRad), separated by SDS-PAGE and analyzed by western blot with the anti-$^N$RTD antibody as described above. The integrity of the YFP tag was confirmed by western blot analysis with 1:5000 anti−GFP polyclonal antibody (Abcam ab6556). Expression tests for each construct were repeated independently at least twice.

To test the ability of aphids to transmit PLRV after exposure to the $^N$RTD via *in planta* expression, *N. benthamiana* leaves were infiltrated with 35:*YFP-$^N$RTD*, 35 S:*$^N$RTD-YFP*, or 35 S:*GFP* (control) constructs. At 2 dpi, aphids were caged on the protein-expressing leaves as well an uninfiltrated (control) leaves for 48 h. Then aphids were moved to PLRV-infected detached HNS leaves for 24 h, and health potato seedlings for 72 h (5 aphids/plant, 10–15 plants/treatment), as in the artificial diet experiments described above. Aphids were removed by a pesticide application and systemic PLRV infection of the potato plants was assessed via DAS-ELISA 2–4 weeks post inoculation. Each inoculated plant represents a replicate (*n* = 37 for all treatments). The experiment was repeated three times independently.

### Transgenic potato delivery of the PLRV $^N$RTD

To generate a plasmid for transformation and expression in potato, the cassette containing 35 S:*YFP-$^N$RTD* from the pEarleyGate104 backbone used for transient expression in *N. benthamiana* was cloned between the ClaI and DraIII restriction sites in pBI121. Western blot analysis (as described above) was used to assess the *in planta* expression of the pBI121 35 S:*YFP-$^N$RTD* construct. This plasmid was transformed into *Agrobacterium tumefaciens* strain AGL1 for potato transformation. Potato plants cv. Desiree were transformed as follows: stem internode segments of 0.5–1 cm in length were excised from six-week-old in vitro-grown plants and incubated with *A. tumefaciens* for 10 min before being moved to callus induction medium. After 48 h, the internode segments were transferred to a selective plant regeneration medium containing kanamyacin. Explants were transferred every 7–14 days to fresh selective plant regeneration medium. When regenerants were approximately 0.5–1 cm in length, they were excised and transferred to a selective rooting medium. Successfully rooted transgenic plants were then transferred to soil.

RT-PCR and confocal microscopy were used to confirm the expression of YFP-$^N$RTD in transgenic potato plants. For RT-PCR, briefly, leaf discs were disrupted in a Mixer Mill 400 (Retsch) and total RNA was extracted using the Zymo Quick-DNA/RNA Miniprep kit. cDNA was synthesized using the iScript Select kit (Bio-Rad) and PCR was carried out using Q5 Hot Start High-Fidelity Polymerase (New

England Biolabs), a YFP-specific forward primer (5′ AAGGGCATCGA CTTCAAGGA 3′) and $^N$RTD-specific reverse primer (5′ ATTGTAGCG TCCCGTTCAAG 3′). RT-PCR analysis was repeated independently at least three times. YFP fluorescence was visualized in epidermal peels using a Leica TCS-SP5 laser scanning confocal microscope running LAS AF software version 2.6.0. YFP was excited with the 514-nm line of a multiline argon laser with emission spectra collected by a hybrid detector (HyD) in the range of 530−569 nm. Chlorophyll was excited by the 514-nm line of a multiline argon laser with emission spectra collected by a HyD in the range of 600−642 nm. All scans were conducted sequentially with line averaging of 8. Empty vector transgenics and wild-type potato plants were imaged with the same settings as controls. The microscopy experiment was repeated independently three times.

To test the effect of the transgenic potato plants on aphid acquisition and transmission of PLRV, *M. persicae* were transferred to YFP-$^N$RTD and empty vector transgenic plants and allowed to feed for 48 h, followed by a 48-hour AAP on PLRV-infected HNS plants. For acquisition, aphids were flash frozen (five aphids/tube, two tubes for empty vector control and two tubes for two independent YFP-$^N$RTD transgenic line) and RNA was extracted and cDNA synthesized as described above. Data for transgenic lines were pooled to calculate standard error for that treatment (Fig. 4c). PLRV titer in aphids was quantified using the QX200 digital droplet PCR system (Bio-Rad)[53]. Briefly, 2 uL of aphid cDNA (normalized to 100 ng/uL) was added to the ddPCR reaction mixture along with 10 uL of 2x EvaGreen SuperMix (Bio-Rad), 7uL of RT-grade $H_2O$, and 0.25 uM of each of the following primers specific to the PLRV CP: FP 5′ TGTCCTTTGTAAACACGA ATGTC 3′ and RP 5′ CTAACAGAGTTCAGCCAGTGG 3′. The reactions were thermocycled and read on the QX200 ddPCR machine according to manufacturer's instructions for EvaGreen SuperMix (Bio-Rad). Droplet counts were analyzed using QuantaSoft (Bio-Rad). For transmission, after the AAP, aphids were moved to uninfected potato plants for a 72-hour IAP. Aphids were devitalized with pymetrozine (Endeavor) and bifenthrin (Talstar P), and systemic PLRV infection of the potato plants was assessed via DAS-ELISA 2–4 weeks post inoculation. Each inoculated plant represents a replicate (*n* = 60 for YFP-$^N$RTD and *n* = 26 for empty vector control). The experiment was repeated twice independently.

### Aphid mortality assays

Mortality of *M. persicae* on PLRV $^N$RTD mutants delivered via artificial diet was assessed using age-synchronized, fourth instar nymphs and adults. Aphids were synchronized by placing adult *M. persicae* aphids on *P. floridana* leaves for two days to lay nymphs. Adults were removed and nymphs were allowed to develop for a week (reaching fourth instar and adulthood) before being used in mortality assays. Purified BSA (*n* = 232), WT PLRV $^N$RTD (*n* = 247), point mutants of the PLRV $^N$RTD H321A (*n* = 183), E366A (*n* = 118), H371A (*n* = 264), E374A (*n* = 161) or one cluster mutant of the PLRV $^N$RTD (containing the three mutations N367, C369, Y410, *n* = 256) were diluted to 0.1 mg/mL in artificial sucrose diet. After starving for 1–2 h, age-synchronized *M. persicae* were placed on artificial diet sachets containing the PLRV $^N$RTD mutants or diet only control for 48 h before being moved to a PLRV-infected or uninfected detached HNS leaf. After 24 h on the HNS leaves, mortality of *M. persicae* was tallied for each treatment and leaf combination. Each individual aphid is considered a replicate. The experiment was repeated three times independently.

### Statistical analysis

PLRV transmission data (Fig. 4, Tables S3, S5, and S7) was analyzed by logistic regression to predict whether an inoculated plant would become infected using the different treatments as predictors. For the transgenic potato transmission experiments (Fig. 4e, d), a one-tailed likelihood ratio test showed that experiment could be removed from

the model as a predictor ($P = 0.7184$, $P = 0.435$, respectively) leaving a single predictor: treatment. For artificial diet and transient *in planta* delivery (Fig. 4a, b), there was a significant effect of experiment (one-tailed likelihood ratio test, $P = 0.002$, $P = 0.0018$, respectively), so a two-predictor model was used: treatment and experiment. This allowed us to observe if there was any effect of treatment on the infection state of the plants after accounting for the variation between experiments. The direction of the effect of a predictor is indicated by the sign of its coefficient (β): a positive coefficient means the predictor increases the chance of a plant becoming infected, negative denotes a decrease. Statistical grouping was determined by comparing the effects (β) of each treatment with one another (one-sided Wald $z$ test for comparison to the control, Wald $\chi^2$ for all other comparisons), using a $P = 0.05$ cut-off. Full models, model diagnostics, test statistics, degrees of freedom, and $P$-values are reported in Tables S4, S6, and S8. PLRV titer in aphids after feeding on transgenic or empty vector control potato plants (Fig. 4d) was compared using an unpaired, one-tailed Student's $t$ test ($P = 0.038$), since the data met the assumptions of normality (Shapiro-Wilk test, $P = 0.142$) and homoscedasticity (Bartlett's test, $P = 0.253$).

Meta-analysis was used to determine the overall effect of aphid pre-exposure to NRTD on PLRV transmission as compared to the no protein control. Subgroup meta-analysis was conducted using the fixed-effects model for between-subgroup-differences using the Mantel-Haenszel method to pool effect sizes and calculate $\tau^2$ since the output data was binary (whether or not a plant became infected with PLRV). The meta-analysis examined all nine artificial diet delivery experiments (pooling the two concentrations used in experiments 1 and 2) and all five *in planta* delivery experiments (transient and transgenic expression). The output of the analysis was used to generate a forest plot which graphically the displays the risk ratio and 95% confidence intervals for each experiment, subgroup, and overall (Fig. S14).

Aphid mortality data (Fig. 5b) was analyzed by quasibinomial regression ($\phi = 0.041$, underdispersed) to predict whether an aphid would die using the different treatments and infection status of the leaf they were moved to (PLRV-infected or uninfected) as predictors. A one-tailed likelihood ratio test showed that experiment could be removed from the model ($P = 0.996$) leaving two predictors: treatment and infection status (Table S10). Statistical grouping was determined by comparing the effects of each treatment with one another (unpaired two-tailed Student's $t$ test), using a $P = 0.05$ cut-off. Full models, model diagnostics, test statistics, and $P$-values are reported in Table S10. All statistical analyses were conducted with R[54] version 3.6.3.

### Reporting summary
Further information on research design is available in the Nature Research Reporting Summary linked to this article.

## Data availability
The atomic coordinates of the TuYV and PLRV NRTD structures are deposited in the Protein Data Bank with accession numbers 7ULN and 7ULO, respectively. The atomic coordinates for the tomato bushy stunt virus coat protein, the turnip yellow virus coat protein, potato leafroll virus coat protein, *S. pneumoniae* PepA, *T. brucei* F1-ATPase, Norovirus Saga GII-4 P domain, and *A. pernix* IF5B that were used for structural comparisons are publicly available from the Protein Data Bank under the accession numbers 2TBV, 6RTK, 6SCO, 3KL9, 65FD, 6H9V, and 5FG3, respectively. All amino acid sequences used for alignment and structural conservation mapping were obtained from the publicly accessible Kyoto Encyclopedia of Genes and Genomes (KEGG) database [https://www.genome.jp/kegg/] (see Fig. S5 for individual sequence IDs). Raw data and images generated in this study associated with PLRV transmission efficiency (Fig. 4) and aphid mortality (Fig. 5) are available in the Source Data file or in

the Supplementary Information. All reagents are available from the corresponding authors upon reasonable request. Source data are provided with this paper.

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

## Acknowledgements

We thank the Northeastern Collaborative Access Team (NE-CAT) beamline staff at the Advanced Photon Source (APS) for assistance with remote X-ray data collection, Dr. Lynn Johnson from the Cornell Statistical Consulting Unit for assistance with statistical methodologies and the meta-analysis of $^N$RTD inhibitor functions, Maria Gutierrez-Feliciano (BTI) for maintenance of plants and insects, and Dr. Mamta Srivastava (BTI) for technical assistance with imaging. We also thank Kerry Swartwood (BTI) for potato transformation and Dr. Mariko Alexander (Cornell) for helping to initiate the collaboration that led to this work. Research conducted at the NE-CAT beamlines (24-ID-C and 24-ID-E) is supported by NIH [P41 GM103403, S10 RR029205] and carried out under the general user proposal GUP-51113 (PI: J.S.C.). This research also used resources of the Advanced Photon Source, a U.S. Department of Energy (DOE) Office of Science User Facility operated for the DOE Office of Science by Argonne National Laboratory under Contract No. DE-AC02-06CH11357. Confocal microscopy images were taken at the BTI Plant Cell Imaging Center, which is supported by the NSF (DBI-0618969). This work was supported by United States Department of Agriculture Grant 2020-67013-31917 (to M.L.H. and J.S.C.) and USDA ARS CRIS Project 8062-22410-007-000-D (to M.L.H.). J.S.C. is a Meinig Family Investigator in the Life Sciences. J.R.W. was supported by a NSF Graduate Research

Fellowship (DGE-1650441) and a NIFA Predoctoral Fellowship (2019-67011-29610). M.C.A. is supported by a NIFA Predoctoral Fellowship (2020-67034-31750). Funding for this research was also provided by the Schmittau Novak Small Grants Program from the Cornell School of Integrative Plant Sciences (to J.R.W. and C.J.S.).

## Author contributions

C.J.S., J.R.W., M.L.H. and J.S.C. designed the study and analysed data. C.J.S., C.J.H., and M.C.A. purified and crystallized all $^N$RTD constructs and collected X-ray diffraction data. C.J.S. solved the $^N$RTD structures and built the models with assistance from M.C.A. and J.S.C. J.S.C. carried out computational modelling. C.J.S. generated and purified all $^N$RTD mutants. M.C.A. performed additional $^N$RTD protein purifications. M.L.H. designed the aphid transmission experiments with J.R.W., H.J.M. and S.E.P. J.R.W. conducted aphid microinjection, artificial diet assays, and transient in planta delivery transmission assays as well as the aphid mortality assays. M.L.H. and J.R.W. designed and generated all expression constructs for transient in planta delivery and potato transformation. J.V.E. supervised the generation of the transgenic potato plants. S.E.P. validated and performed experiments on the transgenic potato plants with microscopy assistance from S.L.D. H.J.M. completed and analysed data for additional transmission assays. J.R.W. and M.L.H. conducted statistical analysis on all aphid experiments. C.J.S., J.R.W., M.L.H., and J.S.C. wrote the manuscript. M.L.H., J.S.C., J.R.W., C.J.S. and M.C.A obtained funding to support the research.

## Competing interests

M.L.H., J.S.C., J.R.W., C.J.S., and M.C.A. have a patent filing related to the technologies described in this paper (U.S. Provisional Patent Application Serial No. 63/289,790). The remaining authors declare no competing interests.
