## [Peer Review File · Nature Communications]

Poliovirus N-terminal readthrough domain structures reveal molecular strategies for mitigating virus transmission by aphidsReviewer #1 (Remarks to the Author):

Schiltz et al provide an ample structural analysis of the read through protein domains (RTDs) associated with turnip yellow virus (TuYV) and potato leafroll virus (PLRV). They provide an attractive description of the evolutionary relationship of the RTDs to the P domains of tombus virus capsids and this insight results in compelling structural models for the location and distribution of the RTDs in the polerovirus, enamovirus, and luteovirus capsids. The structural models of the RTDs and the solubility properties of mutations in the expressed domain provide convincing rationalizations of the phenotypes of the large number of mutants in RTDs reported over many years. Finally, the extended studies on the impact of the expressed RTD domain on aphids (in isolation and in planta), and the unexpected lethality, opens a new area of study for vector control. The only thing lacking is the identification of the aphid gut protein that interacts with the RTD, but that is certainly not necessary for publication of this comprehensive and highly informative investigation of a largely unexplored structural aspect of infection for these groups of viruses.

I found only one issue, below, that needed to be clarified for me. Otherwise, I was satisfied with the presentation and its logic.

1. Line 122 The following sentence is unclear. Is this the case for all the proteins identified by Dali? If so it should be made clear that all of those proteins are related to each other. "The topology differs in TuYV, however, as the individual secondary structure elements are distributed throughout the jellyroll rather than being clustered sequentially as a single globular unit"

Reviewer #2 (Remarks to the Author):

The manuscript « Polerovirus N-terminal readthrough domain structures reveal novel molecular strategies for mitigating virus transmission by aphids », reports the structural characterization of the N-terminal portion of the viral readthrough domain RTD from the poleroviruses turnip yellow virus (TuYV) and potato leafroll virus (PLRV), implicated as a key determinant of aphid transmission. Overall, this is a potentially attractive study that uses x-ray crystallography, mutagenesis and transmission tests, revealing for the first time the atomic structure of the polerovirus N-terminal readthrough domain and a interesting potential way to reduce viral transmission via aphid vectors. Little is know at molecular level regarding interaction between plant viruses and their vectors. The manuscript is clear and well written. Given the experiments performed in this work and the scientific impacts regarding the transmission of plant virus, I believe it is perfectly suited to be published in Nature communications with minor modifications before publication.

Minor remarks :

- It is not clear for me who are corresponding authors.

> Structural organization of polerovirus NRTDs provides insights into evolution of aphid Authors should explain why they decided to keep a so long unfolded c-term sequence. Indeed, the structural model predicted from alphafold reveals a long unfolded C-term part, that is usually removed when trying to over-express a domain with the aim to crystallize it. To be honest, we tried in my team ! we also tried with additionnal 5, 10 and 15 residues in C-ter, always without any expression in bacteria. It is a very interesting case. More than 30 residues predicted as unfolded that are required both for the expression in Bacteria and for maintaining the structure !

-Line 126 : what is the sequence identity of RTD between enamo- and luteoviruses ?

- Beta 11 is missing in figure 1a ??

- In Fig 1b, it would be helpful to indicate the numbers of the residues of secondary elements, and of the C and N ter (230/458).
- Line 187/188 : The triplet deletions : not clear here. As it refers to a supplementary table, the notion of triplet deletions needs to be clarified in the text.
- In Fig 2, it would be easier for the reader if the name of residues was colored with the same code color as the main chain they belong to.
- Authors should indicate in the manuscript the overall dimensions of the RTD dimer, especially to better understand the Fig 3.
- Fig S7 (b,c,d) and FigS8 (a,b,c) as they are zoomed views, we don't know exactly where they are located in the original structure.
- It is not clear for me according to figures : are there some known mutations indirectly involving the C-ter loop ? It is not clear in the manuscript. Considering the fold of this C-ter segment, forming a small beta strand and involved in the dimer interface, the potential role of some mutations needs to be discussed.
- Line 230 : « A composite model combining the TuYV NRTD and CP (PDB: 6RTK) coordinates suggests a similar overall connectivity » : Authors should indicate how many RTD (or how many dimers) are then associated per viral particle according to this stoichiometry.
- Line 264 : « Expression tests and western blot analysis showed that the PLRV NRTD requires a small protein tag to facilitate folding in planta (Fig. S10a-d) ». Does it mean that RTD can not be expressed in planta ?
- In Fig 5 a, it is difficult to know which part of the cap domain we are looking at ! a graphical representation of the area of interest would be helpful.

I am not a specialist of plant virus transmission tests. The vast majority of aphids died after being fed RTD mutants.

- Did you check that mutated RTD are still dimeric in solution ?
- Did you do some transmission tests with survival aphids to know if these aphids are still able to transmit the TUYV, consequently meaning that mutated RTD do not interact with the aphid gut receptor ?
- From a personal point of view, I am not sure that the discovery of a new biopesticide that is lethal for the insect vector is really a good new I will more emphasize on the potential of reducing the interaction of these viruses with their aphid gut receptor that induce mitigated virus transmission by aphids.

Reviewer #3 (Remarks to the Author):

Viruses in the luteo-, polero-, enamo- and luteovirus genera (P/E/L viruses) are transmitted by (and through) aphids in a circulative manner that requires only the virus particle. This virion contains a unique a coat protein (CP) readthrough domain (RTD) which is a long C-terminal extension on a still unquantified subset of the 180 CP subunits in the virion. Ribosomal readthrough of the CP ORF stop codon is required for RTD translation and this efficiency is thought to be on the order of 1-25%. These viruses can be transmitted only by aphids. Thus, a "holy grail" question in P/E/L virology is what is the structure of the RTD on the virion, and what is the structural basis of its interaction with presumed receptors in the aphid that facilitate transmission? The manuscript by Schiltz et al represents a major leap forward toward answering these questions.

This work is also highly significant from a practical agricultural perspective. P/E/L viruses comprise a very diverse and economically important group, featuring the numerous yellow dwarf viruses (in both the polero and luteo genera) of wheat, barley and oats, that rank collectively as the most important viruses of these crops. Given the current worldwide crisis in wheat availability, pervasive yield-reducing diseases such as these viruses are an ever-growing concern.

While the authors did not determine the structure of the RTD (or the full RTP aka CP-RTD) in the context of an intact virion, they did determine at high resolution, the

structures of the RTDs of potato leafroll virus and turnip yellows virus. Given that both the CP domain and the RTD fold into distinct, well-defined structures, it is likely that the RTD structures presented here are essentially the same as those in the context of the virion.

(Because I can't use a superscript in the online review system, I use N-RTD to refer to the <superscript>N RTD notation used by the authors.)

The most striking, unexpected and important finding is that the RTD forms a large 11 beta sheet "jelly roll" domain which bears striking resemblance to the protruding "P" domain in the CP of genus Tombusvirus (Fig. 1). Tombusviruses have much larger CPs than P/E/L viruses, consisting of the S domain which contains the 8 beta sheet jelly roll structure common to all icosahedral virions, and a protruding P domain that is absent in P/E/L viruses. Thus it appears that the RTD may serve the structural role of the P domain and in fact may have been the evolutionary predecessor of the S-P domain arrangement of tombusvirus CPs. In addition to the P domain-like structure, the RTD also contains a novel "cap" domain "on top" of the P domain as drawn in Fig. 1, which the authors propose contributes to the aphid transmission function unique to P/E/L viruses and absent in other tombusvirids. Also different from the CPs of tombusviruses the RTD (and thus its P domain-like structure) is present in less than stoichiometric amounts compared to the rest of the CP, as it is translated by stop codon readthrough. Everything mentioned in this paragraph represents the biggest advance in understanding RTDs since they were genetically determined to be required for aphid transmission in the 1990s.

Another striking, unexpected observation is that certain mutants of the N-RTD cap domain are highly lethal to the aphid. While the mechanism remains unknown, the practical applications of an aphicidal protein are obvious. And the paper leads to exciting questions for future research to determine the mechanism of this lethality.

Overall, the paper is extremely well written and the experimental design and data are displayed clearly and beautifully. Some concerns are discussed below.

Please provide the Genbank accession numbers for the exact isolates of PLRV and TuMV used in this paper, so the reader can determine the precise range of amino acids that define N-RTD. In the Genbank PLRV refseq (NC_00174), the N-terminus of the RTD immediately following the CP stop codon is: VDSGSESPSPQPTPTPQK, with the K being amino acid 230 which the authors report to be the N-terminus of the N-RTD fragment they use in their studies (Suppl Info line 81). Thus, they are not using full N-terminus of the RTD, if my numbering is correct (which is why authors need to provide accession numbers). Technically they are incorrect in implying that the N-RTD is the full N-terminal portion of the RTD, which also looks to be the case in the dashed underlining on supplemental Fig. S1b. I suggest the authors indicate the position of the CP stop codon of each map in Fig. S1b using the same numbering as for the crystallized construct, so the reader can see that the crystallized construct does not start immediately adjacent to the CP ORF stop codon. Moreover, the authors must have added a methionine to allow translation in *E. coli*, which is not present at this position in the RTD. The same applies to TuYV, using accession no. NC_003743, the amino acid 225 at the N-terminus of their N-RTD (suppl info line 82) is also immediately after the alternating proline motif. From what I've read elsewhere, the alternating proline tract omitted by the authors is considered a spacer sequence between the CP domain and the folded, functional portion of the RTD, and I think this alternating proline motif has been shown to be a spacer in other bi-domain proteins, but to my knowledge no one has shown this for these CP-RTD proteins. It would be really interesting to see the structure of this motif in the full CP-N-RTD fusion. However, that might not be crystallizable, and the same may have been the case, if the authors first tried crystallization starting at the exact N terminus (V in above sequence). (If that's the case they should mention it in the paper.) Thus, it is reasonable that the authors omitted this motif, but as far as I can see, the authors do not mention this motif or that their constructs do not reflect the full N-

terminus of RTD anywhere in the manuscript.

In addition, on line 226, the authors mention that an “unstructured linker” connects the S and P domains of TBSV CP, and predict the the same in the region of the C-terminus of CP and N-terminus of the RTD (line 229). They even show it as a yellow dashed line in Fig. 3c, but nowhere do they mention the presence of the alternating proline sequence in this region. Certainly the authors need to explain:

1. that the actual N-terminus of the RTD is not present in their N-RTD proteins.
2. why they did this,
3. why the alternating proline motif is not thought to be a part of the N-RTD fold, and any evidence to support this.
4. and whether the alternating proline motif is thought to act as the unstructured linker or spacer, and does it compare in sequence or (lack of) structure to the same functional motif in TBSV?

Another concern is that all the biological experiments with the N-RTD proteins for both viruses are done with an unnatural amino terminus. The N-RTD protein used here does not exist as such in natural transmissions. It always has the CP fused to it, as well as the connecting alternating proline domain. By having the unnatural N-terminus on all the N-RTD constructs fed to aphids, their half-lives might be much shorter than in nature. More importantly, the N-RTD domain might fold differently in the absence of the upstream sequence. With the entire natural CP-RTD (RTP) protein, it may be difficult to distinguish effects of CP vs RTD on transmission. So, to distinguish the effects of CP from those of RTD, authors could also replace CP with YFP, which they did in one construct, and indeed it was more effective than when N-RTD itself was at the N-terminus (lines 266-268), and the authors point out that this resembles the arrangement with the CP-RTD. This sentence just cries out to have the authors also test the actual CP-RTD protein for comparison. Also, in the YFP-N-RTD construct, did the authors include the putative spacer sequence that includes alternating proline motif? This would make sense to keep YFP fold separate from the RTD fold. While most of the inhibition (or lack thereof) experiments by N-RTD and its mutants do indeed tell us about the effects of the mutations in the context of the 3D structure, it seems obvious and important to have tested the full CP-RTD protein with at least some of the mutations in the RTD. The wild type version of CP-RTD is essential as a positive control. Finally, a lot of the insoluble mutants of the N-RTD might be soluble when expressed fused to the CP?

Also, there is substantial evidence that the RTD is not needed for virions to cross the gut barrier (e.g. Peter et al. (2008), 89, 2037; van den Heuvel (1997) J Virol 71, 7258; Liu et al. (2009) Arch Virol 154,469) yet the authors conclude on line 284 that reduction of aphid transmission by presence of N-RTD is likely due to reducing virions' ability to cross the midgut barrer, without any evidence to distinguish whether the inhibition is due to blocking the gut barrier or the accessory salivary gland barrier, or due to some other mechanism.

The aphid injection experiment (lines 293-301 and Fig. S13) is inconclusive and is missing positive controls. Authors should inject either some kind of inactivated virions that can't infect plants, or perhaps virions from a different polerovirus that is transmitted by *M. persicae* but doesn't infect potato (TuYV?), or at the bare minimum, inject the whole CP-RTD protein, which would provide a much more “natural” stable protein than N-RTD alone, which might be unstable. This leads to the next missing component: there's no evidence of the amount of N-RTD present in the hemocoel after injection, and whether these levels are high enough to compete with virions acquired by feeding. The N-RTD might be quite unstable when injected into the hemocoel. Without those experiments, I recommend deleting this paragraph and associated data. As a result of this, and the previous paragraph, I don't think the authors can conclude anything about the step of transmission at which the RTD is acting from their data. This has been done in previous papers, and can now be pursued in a more informed way in subsequent papers, thanks to the crystal structure presented here.

The structure guided experiments in Fig. 5 and the unexpected lethality of mutants to the aphid is quite interesting. Even though mutant N-RTDs can kill aphids, by an unknown mechanism, this does not reduce the importance of testing constructs with the full CP-RTD to get at the biological mechanisms addressed in the previous experiments.

Typos and other comments on text:

Line 134: they are not aphid transmissible

Line 143: polerovirus virions may be ancestral to tombusvirus virions [Your talking only about the structural proteins. The other viral genes may have other origins.]

Line 248: Orthotospovirus.

Line 304: refer to a figure here. But as mentioned above, I don't see how data identify gut binding by N-RTD, let alone the cap domain.

Line 321: has HNS acronym been defined? nightshade?

Supplementary Information, Lines 139-140: No change needed. Just a comment: It's nice to know that this research was done by real humans who have lives. Rather nice lives it seems, and with a sense of humor!

Manuscript# NCOMMS-22-16721
Response to Referees:

Reviewer #1 (Remarks to the Author):

Schiltz et al provide an ample structural analysis of the read through protein domains (RTDs) associated with turnip yellow virus (TuYV) and potato leafroll virus (PLRV). They provide an attractive description of the evolutionary relationship of the RTDs to the P domains of tombus virus capsids and this insight results in compelling structural models for the location and distribution of the RTDs in the polerovirus, enamovirus, and luteovirus capsids. The structural models of the RTDs and the solubility properties of mutations in the expressed domain provide convincing rationalizations of the phenotypes of the large number of mutants in RTDs reported over many years. Finally, the extended studies on the impact of the expressed RTD domain on aphids (in isolation and in planta), and the unexpected lethality, opens a new area of study for vector control. The only thing lacking is the identification of the aphid gut protein that interacts with the RTD, but that is certainly not necessary for publication of this comprehensive and highly informative investigation of a largely unexplored structural aspect of infection for these groups of viruses.

We thank the reviewer for their positive assessment of our research. We agree the next phase of the research is to identify specific aphid proteins and receptors that bind to the RTD, which will be the research focus of future Heck and Chappie lab members.

I found only one issue, below, that needed to be clarified for me. Otherwise, I was satisfied with the presentation and its logic.

1. Line 122 The following sentence is unclear. Is this the case for all the proteins identified by Dali? If so it should be made clear that all of those proteins are related to each other. "The topology differs in TuYV, however, as the individual secondary structure elements are distributed throughout the jellyroll rather than being clustered sequentially as a single globular unit"

We apologize for the confusion here. The main point we were trying to emphasize is that the TuYV cap domain has a spatially conserved barrel fold that is present in other proteins but is unique in that the secondary structure elements that make up said fold are not connected sequentially but are discontinuous with respect to the amino acid sequence and interspersed throughout the jelly roll domain. The other barrels do in fact show different topologies, but they are all linked sequentially in the sequence (e.g., in Ap IF5B β 25 connects to β 26, which connects to β 27, etc.). We have updated the text as follows to clarify these issues (lines 122-126):

"The DALI alignment algorithm²⁶ indicates that the cap domain barrel is present in several unrelated proteins, including the dimerization domains of aminopeptidases, the N-terminal region of the F1-ATPase rotary subunits, the *Aeropyrum pernix* IF5B initiation factor, and mammalian Norovirus spike proteins (**Fig. S4**). Though the topology differs within these proteins, they each maintain a spatially conserved fold (**Fig. S4b,c**). The TuYV cap domain, however, remains an outlier among this group in that its secondary structure elements are not contiguous, being interspersed throughout the jellyroll domain rather than being connected sequentially to form an isolated globular unit (**Fig. 1b and S4**)."

Reviewer #2 (Remarks to the Author):

The manuscript « Polerovirus N-terminal readthrough domain structures reveal novel molecular strategies for mitigating virustransmission by aphids », reports the structural characterization of the N-terminal portion of the viral readthrough domain RTD from the poleroviruses turnip yellow virus (TuYV) and potato leafroll virus (PLRV), implicated as a key determinant of aphid transmission. Overall, this is a potentially attractive study that uses x-ray crystallography, mutagenesis and transmission tests, revealing for the first time the atomic structure of the polerovirus N-terminal readthrough domain and a interesting potential way to reduce viral transmission via aphid vectors. Little is know at molecular level regarding interaction between plant viruses and their vectors. The manuscript is clear and well written. Given the experiments performed in this work and the scientific impacts regarding the transmission of plant virus, I believe it is perfectly suited to be published in Nature communications with minor modifications before publication.

Minor remarks :

- It is not clear for me who are corresponding authors.

The corresponding authors are Dr. Joshua S. Chappie and Dr. Michelle L. Heck. This is on the cover page with an asterisk and the statement:

“ * To whom correspondence should be addressed. Email: chappie@cornell.edu, mlc68@cornell.edu ”

> Structural organization of polerovirus NRTDs provides insights into evolution of aphid Authors should explain why they decided to keep a so long unfolded c-term sequence. Indeed, the structural model predicted from alphafold reveals a long unfolded C-term part, that is usually removed when trying to over-express a domain with the aim to crystallize it. To be honest, we tried in my team ! we also tried with additionnal 5, 10 and 15 residues in C-ter, always without any expression in bacteria. It is a very interesting case. More than 30 residues predicted as unfolded that are required both for the expression in Bacteria and for maintaining the structure !

The initial ^NRTD constructs were designed years ago before the prominence of AlphaFold. We took a holistic approach that drew from the predictions of multiple fold matching and secondary structure prediction algorithms. Some predictions showed hints of a β -strand with low confidence that was interspersed within regions of high disorder. We used this as a guide to create two constructs. The longer construct remained soluble and yielded the PLRV structure while the shorter construct was completely insoluble. After solving the PLRV structure, we found that the difference between the two constructs was the region constituting the C peptide, with β 17 correlating to the low confidence secondary structure feature present in some predictions. This knowledge allowed us to generate subsequent ^NRTD constructs and ultimately resulted in the TuYV structure. While AlphaFold can be extremely accurate, it is our experience, both here and with several other cases, that it is not infallible and thus it pays to explore other predictive strategies as well.

-Line 126 : what is the sequence identity of RTD between enamo- and luteoviruses ?

We have updated the Data S1 legend to include the percent sequence identity and similarity (as calculated by the Ident and Sim tool in the Sequence Manipulation Suite; Stothard, 2000,

Biotechniques 28:1102-1104) for each viral ^NRTD compared to TuYV as a reference. The reviewer will note that other poliovirus sequences range from ~25%-95% identity, luteoviruses range from 26%-57% identity, and enamoviruses range from ~20-29% identity.

- Beta 11 is missing in figure 1a ??

The labeling in Fig 1a has been updated to include β11. We apologize for this oversight in the initial draft.

- In Fig 1b, it would be helpful to indicate the numbers of the residues of secondary elements, and of the C and N ter (230/458).

While it would be helpful to number each segment, space constraints within the figure would necessitate such a small font that it would be difficult for the reader to decipher these labels. The individual secondary structure elements (e.g., β-sheets, α-helices, and assorted loops) are mapped above the TuYV sequence in the alignment shown in Data S1. We have updated the Fig. 1 legend to refer the reader to this with the following statement:

“See **Data S1** for correspondence between secondary structure elements and the TuYV sequence.”

We have similarly updated the Fig. S3 legend describing the PLRV topology diagram with the following statement:

“See **Data S1** for correspondence between secondary structure elements and the PLRV sequence.”

- Line 187/188 : The triplet deletions : not clear here. As it refers to a supplementary table, the notion of triplet deletions needs to be clarified in the text.

We thank the reviewer for pointing this out. We have clarified this in the text and added an additional mention of Supplementary Fig. S9, which illustrates the location of the specific triplet residue deletions and point mutations in the PLRV ^NRTD structure (Fig. S9a,b). The text now reads (lines 186-191):

“Systematic mutation of conserved residues throughout the PLRV ^NRTD, including a series of triplet residue deletions (**Fig. S9**), previously yielded some mutants where the RTP was not incorporated into the assembled virion and other mutants that were incorporated but were not aphid transmissible¹⁴ (**Table S2**). Many of the mutated side chains are buried and form stabilizing hydrogen bonding and hydrophobic interactions (**Fig. S9a,b**), suggesting that the triplet deletions interfere with the structural integrity and folding of the ^NRTD dimer.”

We have updated the Fig. S9 legend as follows to further aid the reader in connecting the mutated residues with the general structure and the associated phenotypes:

“**Fig. S9. Location and structural stability of PLRV ^NRTD mutants from previous literature.** **a-b**, Location of triplet residue deletions (²⁴¹PML²⁴³, ⁴⁰⁹YNY⁴¹¹, ²³³RFI²³⁵, ²⁶⁸EDE²⁷⁰, and ³¹⁵SST³¹⁷) in the PLRV ^NRTD that produced either unincorporated RTP mutants (**a**) or incorporated RTP mutants that were nontransmissible¹⁸ (**b**). Hydrogen bonds (dashed lines) and neighboring side chains involved in hydrophobic interactions are also shown. **c**, Location of non-

transmissible TuYV point mutants (K404 and Y405)¹⁹. See **Figs. 1, S3, S6, Data S1, and Table S2** for additional info regarding the location and phenotypes of mutated residues in **a-c**.

- In Fig 2, it would be easier for the reader if the name of residues was colored with the same code color as the main chain they belong to.

We appreciate the reviewer's suggestion and have tried throughout the manuscript to match the labeling coloring to the individual subunits/domains/monomers/etc. wherever possible (e.g., Fig. 2a). Given the space limitations of panels 2c-2e, we feel that black provides a better contrast. Match the coloring for residues such as D375^A and E393^B, for example, would require additional shading and/or outlining as it would put the same color font directly against the colored segments of the structure. Although potentially less than ideal, we thus feel this labeling scheme is the clearest and will translate better once the figures are formatted accordingly for publication.

- Authors should indicate in the manuscript the overall dimensions of the RTD dimer, especially to better understand the Fig 3.

We have added a supplementary figure (now Fig. S7) illustrating the dimensions of the TuYV^NRTD dimer and have modified the text in the "C peptide stabilizes the^NRTD dimer interface" section to (lines 159-162):

"Within each dimer, NRTD monomers are oriented parallel to the dimer symmetry axis with the sheet 1 side of the jelly roll facing inward (**Figs. 2a,b and S6a,b**). The dimensions of the TuYV dimer are 63 Å by 62 Å by 42 Å (**Fig. S7**). Cap domain loops L4 and L5 form the upper portion of the TuYV dimer interface..."

- Fig S7 (b,c,d) and FigS8 (a,b,c) as they are zoomed views, we don't know exactly where they are located in the original structure.

Fig. S7a (now Fig. S8a) has been updated with the addition of dashed circles that highlight the regions depicted in the zoomed views in panels b, c, and d. The figure legend for panel a has been updated to read:

"Dashed circles indicate regions highlighted in zoomed views depicted in **b, c, and d**."

Fig. S8 (now Fig. S9) has been updated to include additional labeling of the secondary structure elements to help orient the reader. The legend has also been updated with the following statement to refer the reader to other figures that may help orient them with regard to the positions of the mutated residues:

"See **Figs. 1, S3, S6, and Data S1** for additional info regarding the location of mutated residues in **a-c**."

- It is not clear for me according to figures : are there some known mutations indirectly involving the C-ter loop ? It is not clear in the manuscript. Considering the fold of this C-ter segment, forming a small beta strand and involved in the dimer interface, the potential role of some mutations needs to be discussed.

We refer the reviewer to Fig. S5d,e and the following text on pages 7-8 (lines 167-181):

“The R440 and R443 side chains anchor an extensive network of stabilizing hydrogen bonds and hydrophobic interactions along interior of the structure while β 17 serves a similar role on the exterior (**Figs. 2e and S5d**). Together, the C peptides increase the total buried surface area from 908 Å² to 3615 Å², constituting a major driving force of dimerization. Although we only resolve a partial fragment from one C peptide in the PLRV^NRTD dimer structure (**Figs. S5b and S6a,b**), this piece forms similar stabilizing interactions with both monomers (**Figs. S5d and S6e**). Deletion of the C peptide from either^NRTD expression construct renders the resulting proteins insoluble. ConSurf analysis²⁹ shows that residues directly contacting the TuYV C peptides are highly conserved across all P/E/L viruses (**Fig. S8 and Data S1**), signifying the general importance of these interactions as they are maintained throughout the evolution and adaptation of all three genera. Interestingly, a C-terminal truncation of the CABYV RTP terminating immediately after the C peptide can be efficiently incorporated into mature virions whereas mutants disrupting the anchoring arginines cannot¹³ (**Fig. S5e**). Together these data underscore the critical role the C peptide plays in the proper folding and stability of the^NRTD dimer.”

Updated Figure S5 legend now reads:

“...**d**, Anchoring arginines (R440 and R443 in TuYV, left; R447 and R450 in PLRV, right) stabilize the C peptide through a network of hydrogen bonding interactions (dashed black lines). **e**, C peptide sequences from TuYV, PLRV, and CABYV. Anchoring arginines are marked with dots and highlighted with black boxes. Positions of CABYV C peptide mutants (PCS1 and PCS3+) that prevent RTP incorporation into mature virions¹⁷ are shown below in bold. PCS1 mutant contains alanine substitutions at R441 (anchoring arginine), I442, P443. PCS3+ mutant contains alanine substitutions at R444 (anchoring arginine), V447, and M448 as well as a tryptophan substitution at R445. Arrow denotes the relative position of the CABYV readthrough protein C-terminal truncation that can be normally incorporated into mature virions (RT Δ C_{ter}, residue S462)¹⁷.”

Boissinot et al. (ref 13 in the main text; ref 17 in supplementary materials) showed that only truncations up to what corresponds to the end of the C peptide (denoted in Fig. 5e as RT Δ C_{ter} and marked with an arrow) can be incorporated into mature virions. Additionally, they showed that mutations in the region that corresponds to the C peptide (labeled in Fig. 5e as PCS1 and PCS3+, the nomenclature used in their study) impair the ability of the RTP to be incorporated into virions. In our structures, we find that these mutations primarily disrupt contacts mediated by the anchoring arginines (highlighted in Fig. S5d). Removing these interactions would destabilize the C peptide and likely impair proper folding of the^NRTD, which would explain why the PCS1 and PCS3+ mutants fail to incorporate into virions. We feel this provides additional *in vivo* evidence that supports our structural and biochemical observations and further underscores the importance of the C peptide as the key stabilizing piece required for^NRTD stability.

- Line 230 : « A composite model combining the TuYV NRTD and CP (PDB: 6RTK) coordinates suggests a similar overall connectivity » : Authors should indicate how many RTD (or how many dimers) are then associated per viral particle according to this stoichiometry.

The composite model shown in Fig. 3f illustrates the most likely organization of the^NRTD given the constraints of^NRTD dimerization and the length of the linker connecting the CP to the^NRTD. The structure shown in Fig. 3g is a modeling experiment that illustrates there is no steric clashing if NRTD dimers are positioned around the capsid asymmetric unit. Extrapolating this to the entire capsid would give 90^NRTD dimers in total based on the T=3 symmetry. This fully

occupied stoichiometry has yet to be observed in nature, suggesting that there are other things limits the incorporation of the ^NRTD in infectious virions. We have updated the text to include the total numbers of dimers as requested (lines 239-241):

“This implies that the ^NRTD could feasibly occupy every two-fold position in a poliovirus capsid (yielding a total of 90 ^NRTD dimers) and that the architecture of the ^NRTD itself does not intrinsically limit its stoichiometry.”

- Line 264 : « Expression tests and western blot analysis showed that the PLRV NRTD requires a small protein tag to facilitate folding in planta (Fig. S10a-d) ». Does it mean that RTD can not be expressed in planta ?

The ^NRTD can be expressed *in planta*; however, we have only successfully expressed it with a tag (Fig. S11). An untagged version was not soluble *in planta* and a tag at the C-term does not retain transmission blocking activity (Fig. 4b) and may not retain proper folding. The N-terminal tag facilitated both expression and solubility *in planta* (Fig. S11b) and retained the transmission blocking activity (Fig. 4b).

- In Fig 5 a, it is difficult to know which part of the cap domain we are looking at ! a graphical representation of the area of interest would be helpful.

To help orient the reader, we have added an additional supplementary figure (Fig. S15) that shows positions of the mutated residues in the context of the PLRV ^NRTD dimer in cartoon representation with the secondary structure elements and cap domain loops clearly labeled. This is shown in the same orientation as the Consurf-colored TuYV ^NRTD dimer that is depicted in Fig. 5a. The Fig. 5 legend now also includes the following statement:

“See **Fig. S15** for further representation of where the mutants localize within the PLRV ^NRTD structure.”

I am not a specialist of plant virus transmission tests. The vast majority of aphids died after being fed RTD mutants.

- Did you check that mutated RTD are still dimeric in solution ?

Yes. When analyzed by size-exclusion chromatography (SEC), the purified PLRV ^NRTD mutants used in the feeding assays elute from the Superdex 75 10/300 or Superdex 200 10/300 columns at the same retention volume as the wildtype PLRV ^NRTD, indicating they are soluble, monodispersed, and form the same stable dimers solution. As noted in the methods, SEC via the Superdex 75 is part of the standard purification protocol for all ^NRTD constructs.

- Did you do some transmission tests with survival aphids to know if these aphids are still able to transmit the TUYV, consequently meaning that mutated RTD do not interact with the aphid gut receptor ?

This is a good question. Replicated transmission assays require hundreds of live aphids after acquisition to transfer onto the healthy, recipient plants. We did try these experiments, and it was a challenge to get the number of remaining live aphids at the concentrations shown in this paper. Diluting the concentration of ^NRTD mutants in the diet assays showed reduced mortality to aphids and do block transmission at these lower concentrations (suggesting they more stringently bind the aphid protein regulating virus transmission – but that is yet to be shown), but these data are not included in the paper.

- From a personal point of view, I am not sure that the discovery of a new biopesticide that is lethal for the insect vector is really a good new I will more emphasize on the potential of reducing the interaction of these viruses with their aphid gut receptor that induce mitigated virus transmission by aphids.

This is a very good point by the reviewer, as insecticides alone are not effective at controlling circulative viruses. We modified the sentence slightly to emphasize how the insecticide activity is additive with the transmission blocking potential of the strategy rather than as a stand-alone approach (lines 378-380):

“The deployment of a genetically encoded insecticidal protein that also blocks virus transmission in crop plants lies at the forefront of agricultural technology^{40,41} and may one day eliminate the need for environmentally harmful and costly pesticide applications.”

Reviewer #3 (Remarks to the Author):

Viruses in the luteo-, polero-, enamo- and luteovirus genera (P/E/L viruses) are transmitted by (and through) aphids in a circulative manner that requires only the virus particle. This virion contains a unique a coat protein (CP) readthrough domain (RTD) which is a long C-terminal extension on a still unquantified subset of the 180 CP subunits in the virion. Ribosomal readthrough of the CP ORF stop codon is required for RTD translation and this efficiency is thought to be on the order of 1-25%. These viruses can be transmitted only by aphids. Thus, a “holy grail” question in P/E/L virology is what is the structure of the RTD on the virion, and what is the structural basis of its interaction with presumed receptors in the aphid that facilitate transmission? The manuscript by Schiltz et al represents a major leap forward toward answering these questions.

This work is also highly significant from a practical agricultural perspective. P/E/L viruses comprise a very diverse and economically important group, featuring the numerous yellow dwarf viruses (in both the polero and luteo genera) of wheat, barley and oats, that rank collectively as the most important viruses of these crops. Given the current worldwide crisis in wheat availability, pervasive yield-reducing diseases such as these viruses are an ever-growing concern.

While the authors did not determine the structure of the RTD (or the full RTP aka CP-RTD) in the context of an intact virion, they did determine at high resolution, the structures of the RTDs of potato leafroll virus and turnip yellows virus. Given that both the CP domain and the RTD fold into distinct, well-defined structures, it is likely that the RTD structures presented here are essentially the same as those in the context of the virion.

(Because I can't use a superscript in the online review system, I use N-RTD to refer to the <superscript>N RTD notation used by the authors.)

The most striking, unexpected and important finding is that the RTD forms a large 11 beta sheet “jelly roll” domain which bears striking resemblance to the protruding “P” domain in the CP of genus Tombusvirus (Fig. 1). Tombusviruses have much larger CPs than P/E/L viruses, consisting of the S domain which contains the 8 betasheet jelly roll structure common to all icosahedral virions, and a protruding P domain that is absent in P/E/L viruses. Thus it appears that the RTD may serve the structural role of the P domain and in fact may have been the evolutionary predecessor of the S-P domain arrangement of tombusvirus CPs. In addition to the

P domain-like structure, the RTD also contains a novel “cap” domain “on top” of the P domain as drawn in Fig. 1, which the authors propose contributes to the aphid transmission function unique to P/E/L viruses and absent in other tombusvirids. Also different from the CPs of tombusviruses the RTD (and thus its P domain-like structure) is present in less than stoichiometric amounts compared to the rest of the CP, as it is translated by stop codon readthrough. Everything mentioned in this paragraph represents the biggest advance in understanding RTDs since they were genetically determined to be required for aphid transmission in the 1990s.

Another striking, unexpected observation is that certain mutants of the N-RTD cap domain are highly lethal to the aphid. While the mechanism remains unknown, the practical applications of an aphicidal protein are obvious. And the paper leads to exciting questions for future research to determine the mechanism of this lethality.

Overall, the paper is extremely well written and the experimental design and data are displayed clearly and beautifully. Some concerns are discussed below.

We thank the reviewer for this positive feedback and enthusiasm about our findings.

Please provide the Genbank accession numbers for the exact isolates of PLRV and TuMV used in this paper, so the reader can determine the precise range of amino acids that define N-RTD. In the Genbank PLRV refseq (NC_00174), the N-terminus of the RTD immediately following the CP stop codon is: VDSGSESPSPQPTPTPQK, with the K being amino acid 230 which the authors report to be the N-terminus of the N-RTD fragment they use in their studies (Suppl Info line 81). Thus, they are not using full N-terminus of the RTD, if my numbering is correct (which is why authors need to provide accession numbers).

Data S1 includes the KEGG ID numbers for each sequence used in the alignment. The KEGG database (<https://www.genome.jp/kegg/kegg1d.html>) provides the associated amino acid sequences, the nucleotide sequences, as well as links to other database entries associated with the specific record (e.g., NCBI-GeneID, RefSeq, etc.) where applicable.

Previous studies have used the leaky stop codon as the start point for the readthrough domain. Our work here, however, establishes the ^NRTD as a defined structural unit that begins downstream of this (at residue 230 in PLRV and 224 in TuYV) and is structurally conserved not just between poleroviruses but also with tombusvirus P domains. The intervening sequence between the leaky stop and our structurally defined starting point for the ^NRTD is variable among PEL viruses and is predicted to be unstructured (see further description in responses below). Given our crystallographic observations, we propose to use these definitions moving forward as the domain boundaries of the CP and ^NRTD explicitly, supported by structural and biochemical data.

Technically they are incorrect in implying that the N-RTD is the full N-terminal portion of the RTD, which also looks to be the case in the dashed underlining on supplemental Fig. S1b.

We thank the reviewer for this feedback. Given that we now have a structurally defined unit for the ^NRTD, we propose to keep this definition moving forward and define the intervening region connecting the CP and the ^NRTD as the “variable linker”. We have updated Fig. S1b and Data S1 to reflect these changes and definitions.

I suggest the authors indicate the position of the CP stop codon of each map in Fig. S1b using the same numbering as for the crystallized construct, so the reader can see that the crystallized construct does not start immediately adjacent to the CP ORF stop codon.

We thank the reviewer for pointing this out. We have modified Fig. S1b to emphasize that there is a clear separation between where the CP ends and the folded structure of the ^NRTD begins. The Fig. S1 legend has been amended as follows to reflect these changes:

“...CP and RTD segments are labeled and connected by a variable linker region (dashed line, see also **Data S1**). Domain boundaries of the CP and RTD are derived from structural studies^{17,18}. The relative location of the crystallized ^NRTD construct and C peptide in each RTP are marked. Asterisk denotes position of the leaky stop codon (immediately following 208 in PLRV and 202 in TuYV) that when bypassed yields the full RTP.”

As noted above, the segment that connects these two structurally defined domains is not conserved and predicted to be flexible and unstructured. As such, we propose that the start of the crystallized constructs be used as the defined start point for the RTD moving forward.

We have also updated Data S1 to indicate the position of the variable loop and have updated the Data S1 legend to read:

“...Cap domain loops (L1-5) and C peptide are labeled above along with the variable linker that connects the CP to the ^NRTD (see **Fig. S1b**).”

Moreover, the authors must have added a methionine to allow translation in *E. coli*, which is not present at this position in the RTD. The same applies to TuYV, using accession no. NC_003743, the amino acid 225 at the N-terminus of their N-RTD (suppl info line 82) is also immediately after the alternating proline motif. From what I've read elsewhere, the alternating proline tract omitted by the authors is considered a spacer sequence between the CP domain and the folded, functional portion of the RTD, and I think this alternating proline motif has been shown to be a spacer in other bi-domain proteins, but to my knowledge no one has shown this for these CP-RTD proteins. It would be really interesting to see the structure of this motif in the full CP-N-RTD fusion. However, that might not be crystallizable, and the same may have been the case, if the authors first tried crystallization starting at the exact N terminus (V in above sequence). (If that's the case they should mention it in the paper.) Thus, it is reasonable that the authors omitted this motif, but as far as I can see, the authors do not mention this motif or that their constructs do not reflect the full N-terminus of RTD anywhere in the manuscript.

The methionine added at the start of each ^NRTD should be considered, for all intents and purposes, to be part of the Hrv3C protease site that was present in the NusA fusion construct used to facilitate protein expression in *E. coli*. Including a methionine ahead of an isolated domain in this context is standard practice for all our purification endeavors, not just the ^NRTD constructs. Because the segment that connects the CP and the ^NRTD is not conserved and predicted to be flexible and unstructured, we purposely omitted it in initial ^NRTD construct designs as we anticipated it would hinder crystallization (removing potentially floppy regions is again standard practice). We agree that visualizing this connecting motif by structural methods would be interesting; however, subsequent CP-^NRTD constructs we have investigated after determining the ^NRTD structures all proved to be insoluble in *E. coli*. Attempts to modify/manipulate the linker sequence in the context of the infectious virus would also be interesting but is something that is beyond the scope of this manuscript.

As noted above, we propose that the ^NRTD be defined by the structural domains we present in the paper moving forward rather than the sequence-based definition as in prior publications.

In addition, on line 226, the authors mention that an “unstructured linker” connects the S and P domains of TBSV CP, and predict the the same in the region of the C-terminus of CP and N-terminus of the RTD (line 229). They even show it as a yellow dashed line in Fig. 3c, but nowhere do they mention the presence of the alternating proline sequence in this region. Certainly the authors need to explain:

1. that the actual N-terminus of the RTD is not present in their N-RTD proteins.

Constructs made with the variable linker were not soluble in *E. coli* and this region was truncated in all constructs used for crystallization as it was predicted to be unstructured and flexible.

2. why they did this,

The constructs were attempted but were not soluble when expressed in *E. coli*.

3. why the alternating proline motif is not thought to be a part of the N-RTD fold, and any evidence to support this.

We propose that the ^NRTD be defined by the structural folds we present in the paper, which contain the completely folded P-domain based on homology to tombusviruses.

4. and whether the alternating proline motif is thought to act as the unstructured linker or spacer, and does it compare in sequence or (lack of) structure to the same functional motif in TBSV?

All secondary structure prediction and modeling programs that we have employed (e.g., Phyre, RaptorX, ITASSER, JPRED, GlobPlot, etc.) designate the variable linker region between the CP and ^NRTD as unstructured. We therefore feel confident in modeling it as such, especially given the overall structural homology that is shared with tombusviruses. While the linker region is resolved in the crystal structures of tombusvirus capsids (e.g., PDB: 2TBV, 2ZAH, and 4LLF), we show it as a dashed line in Fig. 3 as we lack coordinates to model it accurately and are inferring its relative position.

Another concern is that all the biological experiments with the N-RTD proteins for both viruses are done with an unnatural amino terminus. The N-RTD protein used here does not exist as such in natural transmissions. It always has the CP fused to it, as well as the connecting alternating proline domain. By having the unnatural N-terminus on all the N-RTD constructs fed to aphids, their half-lives might be much shorter than in nature. More importantly, the N-RTD domain might fold differently in the absence of the upstream sequence. With the entire natural CP-RTD (RTP) protein, it may be difficult to distinguish effects of CP vs RTD on transmission. So, to distinguish the effects of CP from those of RTD, authors could also replace CP with YFP, which they did in one construct, and indeed it was more effective than when N-RTD itself was at the N-terminus (lines 266-268), and the authors point out that this resembles the arrangement with the CP-RTD. This sentence just cries out to have the authors also test the actual CP-RTD protein for comparison. Also, in the YFP-N-RTD construct, did the authors include the putative spacer sequence that includes alternating proline motif? This would make sense to keep YFP fold separate from the RTD fold. While most of the inhibition (or lack thereof) experiments by N-RTD and its mutants do indeed tell us about the effects of the mutations in the context of the 3D

structure, it seems obvious and important to have tested the full CP-RTD protein with at least some of the mutations in the RTD. The wild type version of CP-RTD is essential as a positive control. Finally, a lot of the insoluble mutants of the N-RTD might be soluble when expressed fused to the CP?

We agree with the reviewer, and we did try to make a variety of constructs to express the full-length CP-RTD and CP-^NRTD. Unfortunately, these were not soluble. Although we did not include the alternating proline motif in any of the in vivo constructs, the YFP fusion plasmid encodes a linker (TSLYKKAG) that keeps the protein physically separated from the ^NRTD fold.

Also, there is substantial evidence that the RTD is not needed for virions to cross the gut barrier (e.g. Peter et al. (2008), 89, 2037; van den Heuvel (1997) J Virol 71, 7258; Liu et al. (2009) Arch Virol 154,469) yet the authors conclude on line 284 that reduction of aphid transmission by presence of N-RTD is likely due to reducing virions' ability to cross the midgut barrier, without any evidence to distinguish whether the inhibition is due to blocking the gut barrier or the accessory salivary gland barrier, or due to some other mechanism.

We do not claim in our paper that the ^NRTD in the context of an infectious virion (incorporated) facilitates virus acquisition. Rather, we restrict our conclusions to the evidence we have, which is that the ^NRTD's ability to interfere with virus transmission occurs at the gut barrier. The microinjection experiments of the ^NRTD are expected to bypass the gut barrier and interact at the salivary gland, as numerous papers (including those cited by the reviewer above) show as a means to test for virus interactions at the accessory salivary glands. There was no impact on virus transmission in these experiments, leading us to conclude that the interaction of ^NRTD is at the gut barrier. We do address the alternative explanations proposed by the reviewer in the text of the manuscript (see below), which are certainly plausible. We'd like to draw the reviewer's attention to Reinbold et al., 2001, J. Gen. Vir. (<https://doi.org/10.1099/0022-1317-82-8-1995>), which showed evidence for the ^NRTD in facilitating efficient virus acquisition into the midgut epithelium. Reinbold and colleagues shows that the RTD is not required but it does make the process of virus acquisition into the aphid midgut more efficient. It is also possible that the ^NRTD blocks motifs used by the PLRV CP in virus acquisition in the midgut. Testing this hypothesis is beyond the scope of work in this paper, where we restrict our claims to the evidence showing the ^NRTD transmission blocking mechanism occurs at the gut.

The aphid injection experiment (lines 293-301 and Fig. S13) is inconclusive and is missing positive controls. Authors should inject either some kind of inactivated virions that can't infect plants, or perhaps virions from a different polerovirus that is transmitted by *M. persicae* but doesn't infect potato (TuYV?), or at the bare minimum, inject the whole CP-RTD protein, which would provide a much more "natural" stable protein than N-RTD alone, which might be unstable.

We are unclear as to how injecting inactivated virus that cannot infect plants would serve as a positive control. Inactivated virus would not result in transmission to new plants and we would be unsure as to how to interpret such a negative result. If the reviewer is suggesting that an inactivated virus would also block transmission, that is an interesting idea, but to conclude this we would need to do a thorough analysis on how inactivation changed the structure and whether any of these other viruses intact would block transmission in a manner similar to the ^NRTD. Also, we do not have an infectious clone of TuYV to even perform those experiments readily. Experiments to test whether the purified ^NRTD of one virus species blocks transmission of another are ongoing.

This leads to the next missing component: there's no evidence of the amount of N-RTD present in the hemocoel after injection, and whether these levels are high enough to compete with virions acquired by feeding. The N-RTD might be quite unstable when injected into the hemocoel. Without those experiments, I recommend deleting this paragraph and associated data. As a result of this, and the previous paragraph, I don't think the authors can conclude anything about the step of transmission at which the RTD is acting from their data. This has been done in previous papers, and can now be pursued in a more informed way in subsequent papers, thanks to the crystal structure presented here.

We agree with the reviewer that these are plausible, alternative explanations for failure to see any effect in the microinjection experiments. We added the following text to the end of that paragraph (lines 303-305):

"However, it is possible that the ^NRTD becomes unstable in the hemocoel or is required at a higher concentration than we delivered, which would also explain why no impact on PLRV transmission was observed following delivery by microinjection."

That said, we'd like the reviewer to know we do not think stability of the virus in the hemocoel is the issue. The ^NRTD was extremely stable in a range of conditions in the laboratory, including sitting for weeks at 20°C in a variety of different chemical environments during crystallization and over the course of several days in artificial diet, which is a sucrose solution similar to the osmotic strength found in aphid hemolymph. We injected high concentrations. We were able to see transmission blocking at the gut at 100x dilutions of the concentrations we tested by microinjection.

The structure guided experiments in Fig. 5 and the unexpected lethality of mutants to the aphid is quite interesting. Even though mutant N-RTDs can kill aphids, by an unknown mechanism, this does not reduce the importance of testing constructs with the full CP-RTD to get at the biological mechanisms addressed in the previous experiments.

While we agree with the reviewer, we unfortunately could not perform these experiments as CP-RTD constructs were not soluble.

Typos and other comments on text:
Line 134: they are not aphid transmissible

We corrected the typo.

Line 143: polerovirus virions may be ancestral to tombusvirus virions [Your talking only about the structural proteins. The other viral genes may have other origins.]

We thank the reviewer for bringing this to our attention. We have updated the text to clarify this distinction. The sentence now reads (lines 142-147):

"The conserved topologies between both families (**Fig. 1b and d**) and the intricate distribution of cap domain segments throughout the primary sequence (**Data S1**) suggest that polerovirus structural proteins may be ancestral to those of tombusviruses, with tombusvirus capsids likely evolving via the gradual loss of cap domain elements and truncation of loops L1-L5 rather than through the concerted acquisition of these segments in a manner that would be constrained by the proper folding of both domains."

Line 248: Orthotospovirus.

We corrected the typo.

Line 304: refer to a figure here. But as mentioned above, I don't see how data identify gut binding by N-RTD, let alone the cap domain.

We have altered the text to read (lines 308-312):

“Our modelling suggests that the ^NRTD protrudes from the virion surface with the cap domain directed outward (**Figs. 2a, 3d and 3f**), poised to make direct contact with aphid receptors that contribute to viral transmission¹⁹. We reasoned that mutating surface-exposed side chains within this domain would disrupt critical interactions needed for viral uptake and thus could impair the ability of the ^NRTD to function as an inhibitor in our transmission assays.”

Ref 19: Mulot, M. et al. Transmission of turnip yellows virus by *Myzus persicae* is reduced by feeding aphids on double-stranded RNA targeting the ephrin receptor protein. *Front Microbiol* 9, 457, doi:10.3389/fmicb.2018.00457 (2018).

Previous yeast two hybrid experiments have implicated the *M. persicae* ephrin type-B I-B receptor (Eph) as a potential binding partner for truncated CABYV and TuYV RTP constructs that terminate immediately after the C peptide and dsRNA knockdown of Eph delivered to aphids through feeding reduces TuYV transmission efficiency and inhibits virus uptake (Mulot et al., 2018, *Front. Microbiology*). These observations suggest a direct connection between the ^NRTD, an aphid receptor, and viral uptake and transmission. Our structural modelling – constrained by domain connectivity and the requirement for dimerization – positions ^NRTD on the capsid surface such the cap domain is pointed outward. Given this orientation, we therefore feel it is reasonable to assume that the cap domain would be a main point of contact that could facilitate direct interactions with an aphid receptor and contribute to viral uptake. Moreover, as described above, we have limited our focus here to understanding how the purified ^NRTD can function an inhibitor of transmission. All of this served as the motivation to analyze the effects of cap domain mutations.

Line 321: has HNS acronym been defined? Nightshade?

HNS does stand for hairy nightshade. We have now deleted it here to be consistent, as we did not mention the PLRV source plants in other parts of the results description.

Supplementary Information, Lines 139-140: No change needed. Just a comment: It's nice to know that this research was done by real humans who have lives. Rather nice lives it seems, and with a sense of humor!

We appreciate the reviewer reading the supplementary information until the end. And yes, Carl, the graduate student who solved the structure, was indeed in his beach hotel on vacation in Malta and ran the analysis from his cell phone. Talk about dedication!

Reviewer #2 (Remarks to the Author):

Authors have answered and argued to all the initial requests. I believe it is now suited to be published in Nature communications, with high scientific impacts regarding the transmission of plant virus.

Reviewer #3 (Remarks to the Author):

This is a re-review of the revised ms. See the original review for all my comments about the strength and significance of this paper.

All of my concerns in the previous review have been addressed except for one minor and one minor concern.

The minor concern is my list of 4 explanations I requested regarding line 226 of the original submission on the fact that the authors did not crystallize the fragment or do aphid transmission experiments with the N-RTD fragment starting at the true 5' end of the RTD. The authors adequately responded to me, the reviewer, in their reply, but I meant that they should explain this in the paper to the reader. Some aspects of this are implied in the paper, but it should be spelled out more clearly. Crystallographers get it, more biological virologists and aphid transmission-ologists might be concerned.

My major concern is that Fig. 4e still should be removed from the paper, along with associated text (now lines 295-305 and any other mention of evidence that N-RTD inhibition acts only at the level of gut entry). As stated in my previous review, there's no evidence that any of the proteins, including the N-RTD were correctly injected or accumulated in the hemolymph, let alone at levels to have an effect. In their reply to my previous comment on this, and in the new text they added to the at lines 303-305, they agree that this is plausible. Because it is plausible, the entire experiment is meaningless. Moreover, the conclusion that the N-RTD inhibition occurs at the gut and not salivary glands runs counter to evidence from several other labs that the N-RTD is needed for entry into salivary glands. RTD may also enhance entry at the gut as the authors quote Reinbold et al. 2001, but the data in Fig. 4e don't say anything about this, without better controls, measurement of protein levels. Simply deleting the data and the paragraph is an easy fix for the authors. This reviewer thinks that without these data, the ms. is still acceptable for publication Nature Communications.

Manuscript# NCOMMS-22-16721A

Response to Referees:

Reviewer #2 (Remarks to the Author):

Authors have answered and argued to all the initial requests. I believe it is now suited to be published in Nature communications, with high scientific impacts regarding the transmission of plant virus.

We again thank the reviewer for the helpful comments and suggestions throughout the review process.

Reviewer #3 (Remarks to the Author):

This is a re-review of the revised ms. See the original review for all my comments about the strength and significance of this paper.

We are pleased that the reviewer appreciates the value of our work and the impact that it will have for the field. We thank the reviewer for their insightful comments and useful critiques, which have ultimately improved the manuscript.

All of my concerns in the previous review have been addressed except for one minor and one minor concern.

The minor concern is my list of 4 explanations I requested regarding line 226 of the original submission on the fact that the authors did not crystalize the fragment or do aphid transmission experiments with the N-RTD fragment starting at the true 5' end of the RTD. The authors adequately responded to me, the reviewer, in their reply, but I meant that they should explain this in the paper to the reader. Some aspects of this are implied in the paper, but it should be spelled out more clearly. Crystallographers get it, more biological virologists and aphid transmission-ologists might be concerned.

We value the reviewer's insight here and have thus added a paragraph to the Discussion that provides a brief clarification about nomenclature and domain boundaries (page 14, lines 344-354):

“Previous studies have used the leaky CP stop codon as the starting point for the readthrough domain. Our work here, however, establishes the NRTD as a defined structural unit that begins downstream of this juncture (at residue 230 in PLRV and 224 in TuYV) and is conserved across all PEL viruses (Fig. S5). The intervening sequence that lies between the leaky stop and the beginning of our NRTD crystallographic models is variable and predicted to be unstructured when analyzed by a variety of modelling algorithms (e.g., Phyre, RaptorX, ITASSer, JPRED, GlobPlot, etc.). Given these observations, we propose that the NRTD be defined based on the boundaries elucidated here and that the intervening sequence immediately following the leaky CP stop codon be henceforth referred to as a “variable linker” (Fig. S5). We anticipate these definitions will be more consistent for the field moving forward, especially as we begin to explore the interactions and applications of NRTD constructs deriving from other P/E/L viruses.”

My major concern is that Fig. 4e still should be removed from the paper, along with associated text (now lines 295-305 and any other mention of evidence that N-RTD inhibition acts only at the level of gut entry). As stated in my previous review, there's no evidence that any of the proteins,

including the N-RTD were correctly injected or accumulated in the hemolymph, let alone at levels to have an effect. In their reply to my previous comment on this, and in the new text they added to the at lines 303-305, they agree that this is plausible. Because it is plausible, the entire experiment is meaningless. Moreover, the conclusion that the N-RTD inhibition occurs at the gut and not salivary glands runs counter to evidence from several other labs that the N-RTD is needed for entry into salivary glands. RTD may also enhance entry at the gut as the authors quote Reinbold et al. 2001, but the data in Fig. 4e don't say anything about this, without better controls, measurement of protein levels. Simply deleting the data and the paragraph is an easy fix for the authors. This reviewer thinks that without these data, the ms. is still acceptable for publication Nature Communications.

We have removed the microinjection data from Fig. 4 along with the associated paragraph in the "Results and Discussion" (lines 295-305) and the experimental procedures from the "Methods". We have also removed the "Experimental design for microinjection experiments with the PLRV^NRTD" schematic from the Supplemental Info (formerly Fig. S15).